# ON THE INVERTIBILITY OF INVERTIBLE NEURAL NETWORKS

## ABSTRACT

Guarantees in deep learning are hard to achieve due to the interplay of flexible modeling schemes and complex tasks. Invertible neural networks (INNs), however, provide several mathematical guarantees by design, such as the ability to approximate non-linear diffeomorphisms. One less studied advantage of INNs is that they enable the design of bi-Lipschitz functions. This property has been used implicitly by various works to design generative models, memory-saving gradient computation, regularize classifiers, and solve inverse problems.

In this work, we study Lipschitz constants of invertible architectures in order to investigate guarantees on stability of their inverse and forward mapping. Our analysis reveals that commonly-used INN building blocks can easily become non-invertible, leading to questionable "exact" log likelihood computations and training difficulties. We make use of numerical analysis tools to diagnose non-invertibility in practice. Finally, based on our theoretical analysis, we show how to guarantee numerical invertibility for one of the most common INN architectures.

## 1 INTRODUCTION

Invertible neural networks (INNs) have become a standard building block in the deep learning toolkit. Invertibility is useful for training generative models with exact likelihoods (Dinh et al., 2014; 2017; Kingma & Dhariwal, 2018; Kingma et al., 2016; Behrmann et al., 2019; Chen et al., 2019), increasing posterior flexibility in VAEs (Rezende & Mohamed, 2015; Tomczak & Welling, 2016; Papamakarios et al., 2017), learning transition operators in MCMC samplers (Song et al., 2017; Levy et al., 2017), computing memory-efficient gradients (Gomez et al., 2017; Donahue & Simonyan, 2019), allowing for bi-directional training (Grover et al., 2018), solving inverse problems (Ardizzone et al., 2019) and analysing adversarial robustness (Jacobsen et al., 2019).

The application space of INNs is rapidly growing and many approaches for constructing invertible architectures have been proposed. A common way to construct invertible networks is to use triangular coupling layers (Dinh et al., 2014; 2017; Kingma & Dhariwal, 2018), where dimension partitioning is interleaved with ResNet-type computation. Another approach is to use various forms of masked convolutions, generalizing the dimension partitioning approach of coupling layers (Song et al., 2019; Hoogeboom et al., 2019). To avoid dimension partitioning altogether, multiple approaches based on efficiently estimating the log-determinant of the Jacobian, necessary for applying the change of variable formula, have been proposed to allow for free-form Jacobian structure (Grathwohl et al., 2019; Behrmann et al., 2019; Chen et al., 2019).

From a mathematical perspective, invertible architectures enable several unique guarantees like:

- Enabling flexible approximation of non-linear diffeomorphisms (Rezende & Mohamed, 2015; Dinh et al., 2017; Kingma & Dhariwal, 2018; Chen et al., 2019)
- Memory-saving gradient computation (Gomez et al., 2017; Donahue & Simonyan, 2019)
- Fast analytical invertibility (Dinh et al., 2014)
- Guaranteed preservation of mutual information and exact access to invariants of deep networks (Jacobsen et al., 2018; 2019).

Despite the increased interest in invertible neural networks, little attention has been paid to guarantees on their numerical invertibility. Specifically, this means analyzing their ability to learn bi-Lipschitz

neural networks, i.e. Lipschitz continuous neural networks with a bound on the Lipschitz constant of the forward and inverse mapping.

While the stability analysis of neural networks has received significant attention e.g. due to adversarial examples (Szegedy et al., 2013), the focus here is only on bounding Lipschitz constants of the forward mapping. However, bounding the Lipschitz constant of the inverse mapping is of major interest, e.g. when reconstructing inputs from noisy or imprecise features. In fact, analytical invertibility as provided by some invertible architectures does not necessarily imply numerical invertibility in practice.

In this paper, we first discuss the relevance of controlling the bi-Lipschitz bounds of invertible networks. Afterwards we analyze Lipschitz bounds of commonly used invertible neural network building blocks. Our contributions are:

- We argue for forward and inverse stability analysis as a unified viewpoint on invertible network (non-)invertibility. To this end, we derive Lipschitz bounds of commonly-used invertible building blocks for their forward and inverse maps.
- We numerically monitor and detect (non-)invertibility for different practical tasks such as classification and generative modeling.
- We show how this overlooked issue with non-invertibility can lead to questionable claims when computing exact likelihoods with the change-of-variable formula.
- Finally, we study spectral normalization as a stabilizer for one of the most commonly-used family of INN architectures, namely additive coupling blocks.

## 2 BACKGROUND AND MOTIVATION

Invertible neural networks are bijective functions with a parametrized forward mapping $F_\theta : \mathbb{R}^d \to \mathbb{R}^d$ with $F_\theta : x \mapsto z$, where $\theta \in \mathbb{R}^p$ defines the parameter vector. Additionally, they define an inverse mapping $F_\theta^{-1} : \mathbb{R}^d \to \mathbb{R}^d$ with $F_\theta^{-1} : z \mapsto x$. This inverse can be given in closed-form (*analytical inverse*, e.g. Dinh et al. (2017); Kingma & Dhariwal (2018)) or approximated numerically (*numerical inverse*, e.g. Behrmann et al. (2019); Song et al. (2019)).

Before we discuss building blocks of invertible networks, we provide some background and motivation for studying forward and inverse stability.

**Definition 1** (Lipschitz and bi-Lipschitz continuity). *A function $F : (\mathbb{R}^{d_1}, \|\cdot\|) \to (\mathbb{R}^{d_2}, \|\cdot\|)$ is called* Lipschitz continuous *if there exists a constant $L =: \mathrm{Lip}(F)$ such that*

$$\|F(x_1) - F(x_2)\| \leq L\|x_1 - x_2\|, \quad \forall x_1, x_2 \in \mathbb{R}^{d_1}.$$

*If an inverse $F^{-1} : (\mathbb{R}^{d_2}, \|\cdot\|) \to (\mathbb{R}^{d_1}, \|\cdot\|)$ and a constant $L^* =: \mathrm{Lip}(F^{-1})$ exists such that*

$$\|F^{-1}(y_1) - F^{-1}(y_2)\| \leq L^*\|y_1 - y_2\|, \quad \forall y_1, y_2 \in \mathbb{R}^{d_2},$$

*then $F$ is called* bi-Lipschitz continuous.

**Remark 2.** *We focus on invertible functions $F : (\mathbb{R}^d, \|\cdot\|_2) \to (\mathbb{R}^d, \|\cdot\|_2)$, i.e. functions where the domain and co-domain are of the same dimensionality $d$ and the norm is given by the euclidian norm.*

**Lemma 3.** *(Rademacher (Federer, 1969, Theorem 3.1.6))*
*If $F : \mathbb{R}^d \to \mathbb{R}^d$ is a locally Lipschitz continuous function (i.e. functions whose restriction to a neighborhood around any point is Lipschitz), then $F$ is differentiable almost everywhere. Moreover, if $F$ is Lipschitz continuous, then*

$$\mathrm{Lip}(F) = \sup_{x \in \mathbb{R}^d} \|J_F(x)\|_2,$$

*where $J_F(x)$ is the Jacobian matrix of $F$ at $x$ and $\|J_F(x)\|_2$ denotes its spectral norm.*

Lipschitz bounds on the forward mapping are of crucial importance in several areas, including in adversarial example research (Szegedy et al., 2013), to avoid exploding gradients, or the training of Wasserstein GANs (Anil et al., 2019). The stability of the inverse, however, can have a similar impact. For instance, having a Lipschitz bound on the inverse may avoid vanishing gradients during training.

Given that deep-learning computations are carried out with limited precision, imprecision is always introduced in both the forward and backward passes, i.e., $z^\delta = F(x) + \delta$ and $\hat{x}^\delta = F^{-1}(z^\delta)$. Instability in either pass will aggravate this problem, and essentially make the invertible network numerically non-invertible. To summarize, this problem occurs in the following situations:

- Numerical reconstruction of $x$, where features $z^\delta$ are inexact due to limited precision (e.g. when computations are executed in single precision as common on modern hardware).
- Reconstruction based on imprecise measurements from physical devices (e.g. when using invertible networks for inverse problems (Ardizzone et al., 2019)).
- Numerical re-computation of intermediate activations of the neural network to allow for memory-efficient backpropagation (Gomez et al., 2017).

Furthermore, some computations are performed via numerical approximation, which in turn adds another source of imprecision that might be aggravated via instability. Examples include:

- Numerical forward computation, as in Neural ODEs (Chen et al., 2018) (numerical solver is used to approximate dynamic of ODE).
- Numerical inverse computation, e.g. via fixed-point iterations as in i-ResNets (Behrmann et al., 2019) or MintNet (Song et al., 2019) or via ODE-solvers for the backward dynamics as in Neural ODEs (Chen et al., 2018).

As an example of why bi-Lipschitz continuity is critical for numerical stability in invertible functions, let's consider the simple mappings $F_1(x) = \log(x)$, $F_1^{-1}(z) = \exp(z)$, and $F_2(x) = x, F_2^{-1}(z) = z$. Though both functions tend to infinity when $x \to \infty$, $F_1$ is much less stable. Consider the introduction of numerical imprecision as $z^\delta = F_1(x) + \delta$ where $\delta$ denotes the introduced imprecision. Then this imprecision is magnified in the inverse pass as:

$$||F_1^{-1}(z) - F_1^{-1}(z^\delta)||_2^2 \approx ||\delta \frac{\partial F_1^{-1}(z^\delta)}{\partial z^\delta}||_2^2 = ||\delta \exp(z^\delta)||_2^2. \qquad (1)$$

A similar example can be constructed for both the forward and backward passes, which speaks to the importance of bi-Lipschitz continuity. For an additional discussion on the connection of Lipschitz constants and numerical errors, we refer to Appendix B.

## 3 STABILITY OF INVERTIBLE NEURAL NETWORKS

### 3.1 LIPSCHITZ BOUNDS FOR BUILDING BLOCKS OF INVERTIBLE NETWORKS

Research on invertible networks has produced a large variety of architectural building blocks. Yet, the focus of prior work was on obtaining flexible architectures while maintaining invertibility guarantees. Here, we build on the work in (Behrmann et al., 2019), where bi-Lipschitz bounds were proven for invertible ResNets, by deriving Lipschitz bounds on the forward and inverse mapping of common building blocks. Together with an overview of common invertible building blocks, we provide our main results in Table 1. We chose these particular model classes in order to cover both coupling-based approaches and free-form approaches like Neural ODE (Chen et al., 2018) and i-ResNets (Behrmann et al., 2019). The derivations of the bounds are given in Appendix A. Note that the bounds provide the worst-case stability and serve mainly as a guideline for future designs of invertible building blocks.

### 3.2 CONTROLLING STABILITY OF BUILDING BLOCKS

As shown in Table 1, there are many factors that influence the stability of INNs. Of particular importance are the Lipschitz constants $\text{Lip}(g)$ of the sub-network $g$ for i-ResNets (Behrmann et al., 2019) and affine coupling blocks (Dinh et al., 2014), and $\text{Lip}(s)$, $\text{Lip}(t)$ for additive coupling blocks (Dinh et al., 2017). Whereas computing the Lipschitz constants of neural networks is NP-hard (Virmaux & Scaman, 2018), there is a simple data-independent upper bound:

$$\text{Lip}(g) \le \prod_{i=1}^{L} \|A_i\|_2, \quad \text{for} \quad g(x) = A_L \circ \phi \circ A_{L-1} \circ \cdots \circ A_2 \circ \phi \circ A_1, \qquad (2)$$

| Building Block | Forward Operation | Lipschitz Forward | Lipschitz Inverse |
|---|---|---|---|
| **Additive Coupling Block** (Dinh et al., 2014) | $F(x)_{I_1} = x_{I_1}$ $F(x)_{I_2} = x_{I_2} + g(x_{I_1})$ | $\leq 1 + \mathrm{Lip}(g)$ | $\leq 1 + \mathrm{Lip}(g)$ |
| **Affine Coupling Block** (Dinh et al., 2017) | $F(x)_{I_1} = x_{I_1}$ $F(x)_{I_2} = x_{I_2} \odot g(s(x_{I_1})) + t(x_{I_1})$ $g(\cdot) \neq 0$ | $\leq \max(1, c_g) + M$ local for $x \in [a,b]^d$ $g(x) \leq c_g$ | $\leq \max(1, c_{\frac{1}{g}}) + M^*$ local for $y \in [a^*, b^*]^d$ $\frac{1}{g}(y) \leq c_{\frac{1}{g}}$ |
| **Invertible Residual Layer** (Behrmann et al., 2019) | $F(x) = x + g(x)$ $\mathrm{Lip}(g) < 1$ | $\leq 1 + \mathrm{Lip}(g)$ | $\leq \frac{1}{1 - \mathrm{Lip}(g)}$ |
| **Neural ODE** (Chen et al., 2018) | $\frac{dx(t)}{dt} = F(x(t), t)$ $t \in [0, T]$ | $\leq e^{\mathrm{Lip}(F) \cdot t}$ | $\leq e^{\mathrm{Lip}(F) \cdot t}$ |
| **Invertible Downsampling / Squeeze** (Dinh et al., 2017) | $F(x) = Px$ $P$ permutation | $= 1$ | $= 1$ |
| **Diagonal Scaling** (Dinh et al., 2014) **ActNorm** (Kingma & Dhariwal, 2018) | $F(x) = Dx$ $D$ diagonal $D_{ii} \neq 0$ | $= \max_i |D_{ii}|$ | $= \frac{1}{\min_i |D_{ii}|}$ |
| **Invertible** $1 \times 1$ **Convolution** (Kingma & Dhariwal, 2018) | $F(x) = PL(U + \mathrm{diag}(s)) =: W$ $P$ permutation, $L$ lower-triangular $U$ upper-triangular, $s \in \mathbb{R}^d$ | $\leq \|W\|_2$ | $\leq \|W^{-1}\|_2$ |

Table 1: **Lipschitz bounds on building blocks of invertible neural networks.** The second column shows the operations of the forward mapping and the last two columns show bounds on the Lipschitz constant of the forward and inverse mapping. $M$ in the row for the forward mapping of an affine block is defined as $M = \max(|a|, |b|) \cdot c_{g'} \cdot \mathrm{Lip}(s) + \mathrm{Lip}(t)$. Furthermore, $M^*$ for the inverse of an affine block is $M^* = \max(|a^*|, |b^*|) \cdot c_{(\frac{1}{g})'} \cdot \mathrm{Lip}(s) + c_{(\frac{1}{g})'} \cdot \mathrm{Lip}(s) \cdot c_t + c_{\frac{1}{g}} \cdot \mathrm{Lip}(t)$. Note that the bounds of the affine blocks hold only locally. Derivations of the bounds are given in Appendix A.

where $A_i$ are linear layers, $\| \cdot \|_2$ is the spectral norm and $\phi$ a contractive activation function ($\mathrm{Lip}(\phi) \leq 1$). The above bound was used by (Behrmann et al., 2019) in conjunction with spectral normalization (Miyato et al., 2018; Gouk et al., 2018) to ensure a contractive residual block $g$. In particular, this employs a normalization via:

$$\tilde{A} = \kappa \frac{A}{\hat{\sigma}_1}, \quad \text{with} \quad \hat{\sigma}_1 \approx \sigma_1 = \|A\|_2 \quad \text{(approx. via power-method)},$$

where $\kappa > 0$ is a coefficient that sets the approximate upper bound on the spectral norm of each linear layer $A_i$. Thus, by setting an appropriate coefficient $\kappa$ depending on the targeted Lipschitz bound of the building block, this approach enables one to control both forward and inverse stability. Note that the above discussion can be generalized to other $\ell_p$-norms, see (Chen et al., 2019).

However, this is not sufficient when using affine coupling blocks because their bound on the Lipschitz constant holds only locally. In particular, it depends on the regions of the inputs $x$ to the coupling block. While inputs to the first layer are usually bounded by the nature of the data, obtaining bounds for intermediate activations is less straightforward. One interesting avenue for future work could be local regularizers like gradient penalties (Gulrajani et al., 2017), where spectral normalization could be used post-hoc to certify stability.

Lastly, we use ActNorm (Kingma & Dhariwal, 2018) in several architectures and avoid small diagonal terms which would yield large Lipschitz constants in the inverse (see Table 1) by adding a positive constant. Further stabilization could be achieved via bounding the scaling.

## 4 NUMERICAL EXPERIMENTS

In this section, we study numerical invertibility for several objectives and architecture settings. This section is structured by task:

1. **Classification:** we show that INN classifiers can become non-invertible on CIFAR-10 and discuss consequences for memory-efficient backpropagation.
2. **Density estimation:** we analyze the numerical invertibility of SOTA trained density models.
3. **Generative modeling:** we study the stability of adversarially trained INN generators, discuss the consequences for likelihood evaluation, and stabilize an additive-coupling based INN generator using spectral normalization.
4. **Decorrelation:** we perform an in-depth study of the effect of different architecture settings on a simple task, where both stable and unstable solutions are possible. Furthermore, we show that spectral normalization is effective at stabilizing additive-coupling based flows.

We use the following measures to diagnose the numerical instability of invertible models:

- **Reconstruction error.** We measure the $\ell_2$-distance between the input $x$ and its reconstruction, i.e. $||x^{(i)} - F_\theta^{-1}(F_\theta(x^{(i)}))||_2$.
- **Conditioning of the Jacobian and max/min singular values.** For forward stability, we are interested in the behavior of the Jacobian $J_F(x)$, while for inverse stability we are interested in the Jacobian of the inverse mapping $J_{F^{-1}}(x)$. We compute the singular values of the Jacobians using the SVD, which allows us to compute its condition number. [1]

While the reconstruction error allows us to quantitatively monitor non-invertibility even before reconstruction artifacts are perceptible, the linear approximation $J_F(x)$ and its singular values provide insights into unstable directions of the forward (very large singular values of $J_F(x)$) and inverse (very small singular values of $J_F(x)$) mapping. Both measures were also used in (Jacobsen et al., 2018), where an ill-conditioned inverse was observed.

## 4.1 CLASSIFICATION WITH INVERTIBLE MODELS

In this section, we show that when training an INN for classification, there exist both stable and unstable solutions. We compare a stable model that uses additive coupling and an unstable model that uses affine coupling and ActNorm both inside and between the blocks—for additional experimental details see Appendix C. These models achieve similar test accuracies of 90.2% and 90.5%, respectively (Figure 6, Appendix C); however, we note that the goal of this experiment is not to achieve SOTA accuracy on CIFAR-10. Rather, we aim to show that models trained for classification with reasonable accuracy, can vary greatly with respect to stability. To observe the differences in stability, we plot the reconstruction results in Figure 2.

An important use-case of INNs is to enable memory-efficient training, by re-computing activations in the backward pass rather than storing them in memory during the forward pass (Gomez et al., 2017). This approach enables e.g. large-scale generative modeling (Donahue & Simonyan, 2019) and scaling segmentation networks to high-resolution medical images (Brügger et al., 2019).

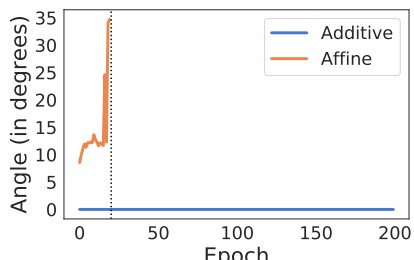

Figure 1: **Impact of stability on memory-efficient training.** We measure the angle between the true gradient (using stored activations) and the memory-saving gradient (using re-computing activations). For all epochs after the dotted vertical line, the affine model had numerically infinite or `nan` gradients, while the additive model gives accurate memory-efficient gradients.

Re-computing the activations, however, relies on a numerically precise inverse mapping. To better understand the effect of numerical errors, we perform an analysis similar to Gomez et al. (2017): we track the angle between the true and memory-saving gradients during training (Figure 1). As expected from the reconstruction results, we observe that the affine model yields gradients very different from the true gradient; in fact, after approximately 20 epochs of training, the memory-saving gradients of the affine model contain numerically infinite or `nan` values. Thus, it would not be possible to train the affine model successfully using memory-saving gradients.

[1] Using the SVD is feasible for CIFAR-10, but becomes prohibitively expensive for ImageNet (with images of size $3 \times 256 \times 256$); for such larger images, one could instead use the Lanczos algorithm (Lanczos, 1950) to find the largest and smallest singular values, to compute the condition number.

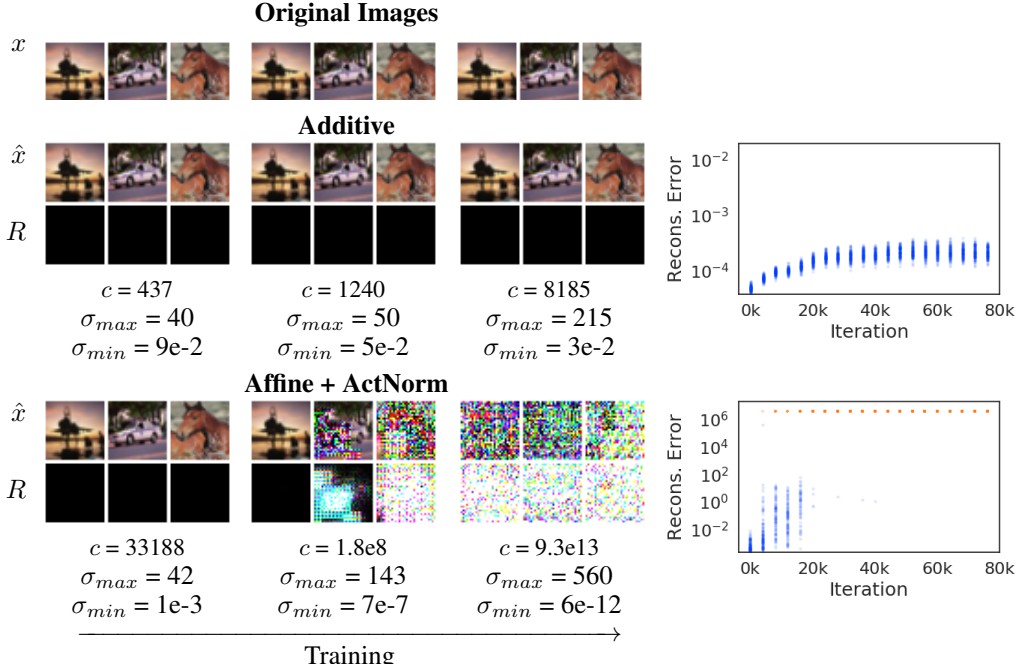

Figure 2: **Measuring stability of additive and affine INN classifiers on CIFAR-10.** Right: we track the reconstruction errors for a fixed minibatch over the course of training. In each iteration (vertical slice) we plot the errors of each minibatch element. Orange points in the affine plot denote `inf` and `nan` values. $c$ denotes the condition number of the Jacobian, $\sigma_{max}$ and $\sigma_{min}$ denote the max and min singular values. At the top, we show the original images $x$ and in each snapshot, the reconstructions $\hat{x} = F^{-1}(F(x))$ and reconstruction errors $R = |x - \hat{x}|$.

## 4.2 ANALYZING STATE-OF-THE-ART DENSITY MODELS

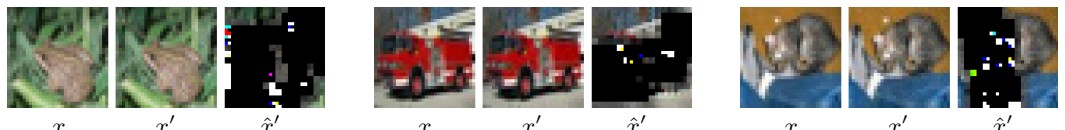

Figure 3: **Crafting non-invertible inputs for a CIFAR-10 Glow model.** For three different images, we show: 1) the original datapoint $x$ from which we start running PGD; 2) the crafted input $x'$ that results from PGD; and 3) the reconstruction $\hat{x}' = F^{-1}(F(x'))$ of the crafted input. In all cases, the reconstructions of adversarial inputs are heavily corrupted, indicating that the attacks were successful, and exposing non-invertibility in this Glow model.

In this section, we analyze the invertibility of trained density models. In particular, we expose non-invertibility in models that otherwise appear stable by optimizing in input space to find examples that are poorly reconstructed by the model. Here, we take a trained Glow model (Kingma & Dhariwal, 2018)[2] and optimize the input using Projected Gradient Descent (PGD) (Madry et al., 2018). Our goal is to find a point $x'$ in the domain of the invertible model such that the reconstruction $F^{-1}(F(x'))$ differs from $x'$. In particular, we start with a datapoint $x$ and use PGD to find a perturbed example $x'$ that has high reconstruction error via Eq. 3 (additional details in Appendix E).

$$\underset{||x'-x||_\infty \leq \epsilon}{\arg\max} \ ||x' - F^{-1}(F(x'))||_2. \tag{3}$$

As shown in Figure 3, this attack is effective for finding examples that are perceptually identical to test examples, yet induce large reconstruction errors. This attack can be understood as a worst-case invertibility diagnosis; however, we note that unsuccessful attacks can be due to algorithmic

---

[2]We used the PyTorch implementation from `https://github.com/y0ast/Glow-PyTorch`. This trained model achieves a likelihood of 3.39 bits-per-dimension.

issues and thus do not necessarily imply stable invertible models. Also, it is not always clear how to get gradients that are able to exploit numerical instabilities, leaving room for improvement via gradient-free methods (e.g., the boundary attack from Brendel et al. (2018)).

We present in Appendix E additional experiments using the PGD attack on the additive Glow model from (Kingma & Dhariwal, 2018) trained on CelebA, where we show that the model becomes non-invertible outside the valid input range of images. Furthermore, we analyze a trained residual flow model (Chen et al., 2019) on CIFAR-10, where we cannot find such dramatic non-invertible inputs, which is to be expected given the stability bounds of i-ResNet blocks (see Table 1).

### 4.3 GENERATIVE MODELING WITH INVERTIBLE MODELS

Generative models based on invertible networks have been predominantly trained using maximum likelihood estimation (MLE). Another viable approach is to train them adversarially (ADV), as done in Flow-GAN (Danihelka et al., 2017; Grover et al., 2018). Flow-GAN is appealing as it can result in a generator capable of producing high-quality samples (as in GANs), while also giving access to exact density estimates, which GANs lack. Prior work (Danihelka et al., 2017; Grover et al., 2018) has compared these two techniques for training flows (MLE vs ADV); the main conclusion of these studies was that training with MLE yields good likelihoods but relatively poor samples, while training with a GAN loss yields good samples but *likelihoods orders of magnitude worse than MLE training*.

|  | **MLE** Affine | **MLE** Additive | **ADV** Additive | **ADV** Additive |
|---|---|---|---|---|
|  | - | - | **Unstable** | **Stable** |
| (Test) BPD | 1.22 | 1.38 | 3.87e12* | 709 |
| FID | 99.9 | 95.1 | 25.3 | 195 |

Table 2: Comparison of bits-per-dimension (BPD) and sample quality via FID scores for MLE- and ADV-trained models. For reference, the FID of a untrained Flow is roughly 1500. Using the stable version with ADV, though improves BPD significantly, might come at a trade-off on FID.

Here, we analyze in depth the effect of Flow-GAN training on numerical stability. We use networks with repeated **additive coupling** layers, and **ActNorm** between blocks. We examine two architectures, both with 3 levels, i.e., 'squeeze' between levels (additional details in Appendix D):

- **Stable**: having a depth of 4 (i.e., 4 blocks per level) and spectral normalization applied to all the convolution layers, see section 3.2 for details.
- **Unstable**: having a depth of 16, ActNorm within coupling layer, and no spectral normalization applied to the convolutional layers.

Table 2 shows that models trained with MLE objective can achieve good bits-per-dimension (BPD), but models trained with ADV can achieve better sample quality as measured by the Frechet Inception Distance (FID), a common measure of sample quality (Heusel et al., 2017; Lucic et al., 2018). This justifies why considering training INN with objective functions other than MLE is desired.

**Broken Flow-GAN.** In Figure 4, we show that an INN trained only with adversarial loss can become non-invertible, depending on the architecture. We perform forward and inverse passes repeatedly on the same mini-batch. The unstable model shows visible reconstruction errors quickly. Table 3 shows that the unstable model has BPD orders of magnitude larger than the stable model.

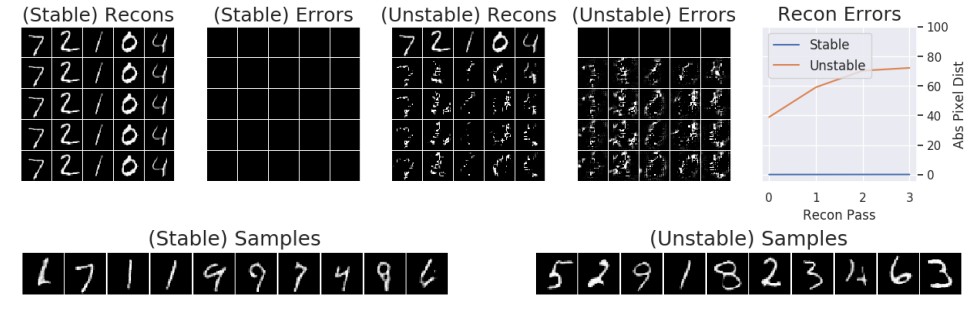

Figure 4: Reconstructions of images through multiple forward/inverse passes. First row in the 'Recons' subfigures are real-data, and each row after is one forward/inverse pass.

| Model | (Test) BPD | Max-SV | Min-SV | Cond-Num | Analytic LDJ |
|---|---|---|---|---|---|
| **ADV (Stable)** | 709 | 1.99e3 | 0.6129 | 3.253e3 | 4734 |
| **ADV (Unstable)** | 3.87e12* | 3.83e8 | 0.0309 | 1.239e10 | 9874 |

Table 3: Stability analysis on two architectures trained adversarially (ADV), and with maximum likelihood (MLE). *means this number is meaningless as the network is visibly non-invertible. LDJ denotes the log-determinant of the Jacobian, BPD denotes bits-per-dimension and SV singular value.

**Flow-GAN "Likelihood".** Typically likelihood is computed by a change of variables, which assumes invertibility. When a network is numerically non-invertible, the assumption breaks, and the computed value becomes some numerical approximation to the density. The model used in Figure 4 consists of additive coupling blocks, ActNorm, and squeezing operations. All these operations have *data independent* log-determinant Jacobian. Thus, obtaining a numerical value via the change of variable formula is straightforward. However, it is unclear what this numerical value represents, likely it cannot be trusted as true likelihood due to the lack of invertibility. In this case, we advocate for ensuring invertibility using the remedies discussed here to make sure BPD values are trustworthy. In terms of sample quality, however, we currently observe a tradeoff since the stable model yields higher FID scores than the unstable model (Table 2, includes also MLE-models for comparison). Yet, we believe that proper tuning e.g. of spectral normalization could remove this tradeoff.

Lastly, one can adjust for the likelihood by adjusting the prior (see Appendix I). In summary, in this section we point out that Flow-GAN can become non-invertible, in which case the computed likelihood cannot be taken as ground truth likelihood (Grover et al., 2018). In sum, INN trained with MLE are stable, but the sample quality is worse than those trained with ADV. Yet, training with ADV loss might make INN non-invertible, which defeats the purpose of using INN in the first place. Hence, when training with alternative objectives, numerical stability is a crucial property to consider.

## 4.4 DECORRELATION TASK

As the last part of our empirical study, we use a simple decorrelation task to benchmark the stability of invertible models. In particular, we compute the correlation matrix $C \in \mathbb{R}^{d \times d}$ via

$$C_{j,l}^{\theta} = \frac{1}{N} \sum_{i=1}^{N} \frac{\left( F_{\theta}(x^{(i)})_j - \hat{\mu}_j \right) \left( F_{\theta}(x^{(i)})_l - \hat{\mu}_l \right)}{\hat{\sigma}_j \hat{\sigma}_l}, \tag{4}$$

where $\hat{\mu}_j$ is the estimated mean over output samples $F_{\theta}(x^{(i)})$ and $\hat{\sigma}_j$ the estimated standard deviation. Then, we optimize the parameters $\theta$ to minimize the off-diagonal correlation, i.e.

$$\min_{\theta} \| C^{\theta} - \mathrm{diag}(C^{\theta}) \|_F, \tag{5}$$

where $\| \cdot \|_F$ is the Frobenius-norm [3]. Decorrelation objectives have been used in (Cogswell et al., 2015) to reduce overfitting and in (Cheung et al., 2014) to disentangle hidden activations.

This objective serves as a good task for our purposes for two reasons:

1. Decorrelation is a simpler objective than optimizing outputs $z = F_{\theta}(x)$ to follow a factorized Gaussian as in Normalizing Flows (Rezende & Mohamed, 2015). Furthermore, it will show that changing the objective to less standard tasks can lead to larger instabilities compared to using INNs on more common tasks such as density estimation.
2. Decorrelation allows multiple solutions using invertible mappings, where both stable and unstable transforms are equally valid for the given objective. See Appendix F for a motivation based on a simple 2D toy example.

In summary, the decorrelation objective offers an environment to study which INN components steer the mapping towards stable or unstable solutions, that are equally plausible for the given task.

In our experiments shown in Figure 5, we focus on coupling-based models like Glow (Kingma & Dhariwal, 2018), which are analytically invertible and thus allow to a simpler analysis compared to models relying on numerical inversion like i-ResNets (Behrmann et al., 2019). We evaluate the effects of different architectural choices on numerical stability, including additive vs. affine coupling layers, ActNorm, and architecture depth. For ActNorm we study two settings: 1) between coupling blocks, 2) inside blocks, i.e. as part of the function $g$ in additive blocks or $s$ and $t$ in affine blocks. Details on the architectures/ training schemes and extended results are provided in Appendix H.

---

[3]We provide example PyTorch code for the decorrelation objective in Appendix G.

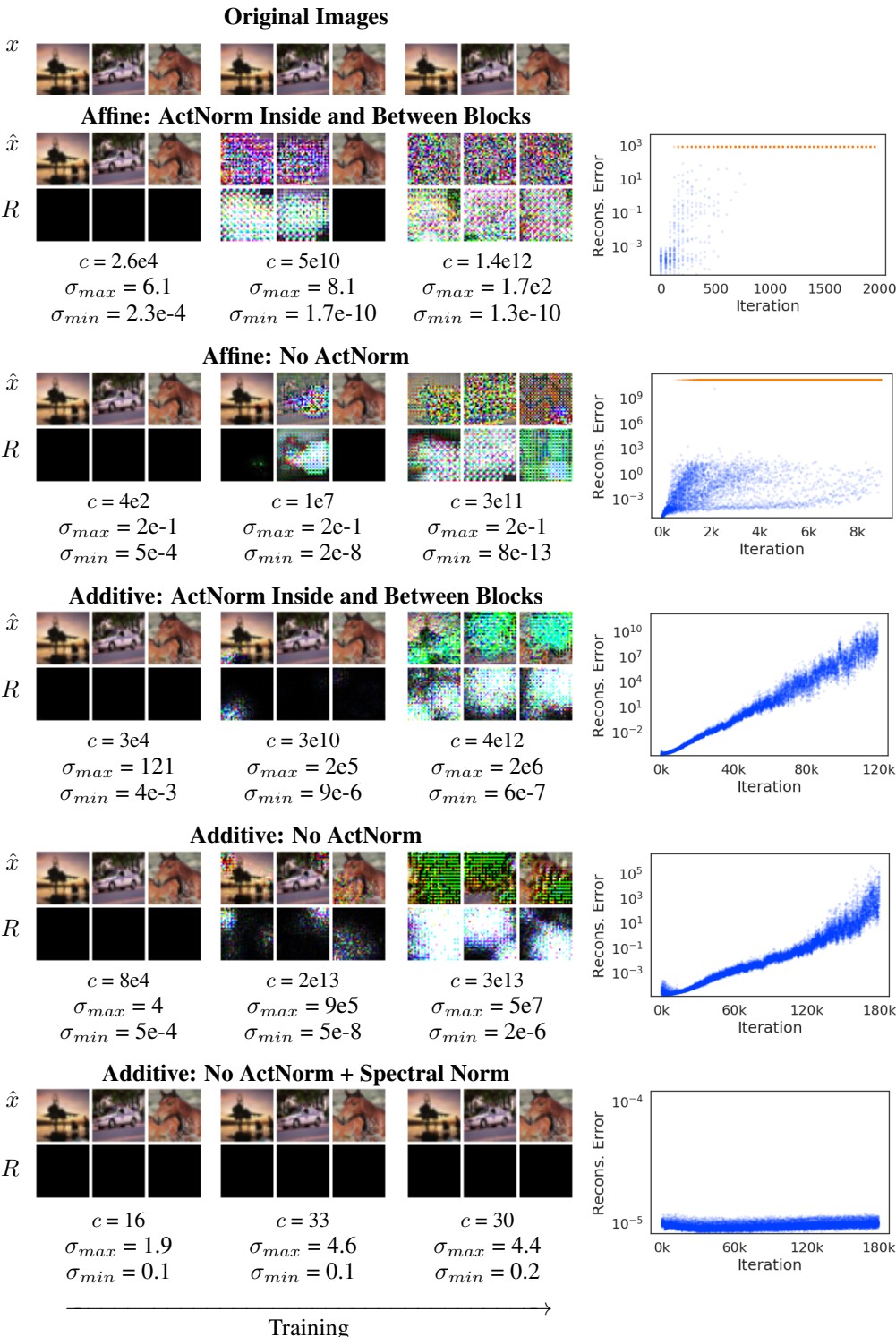

Figure 5: **Instability in affine and additive models.** On the right, we track the reconstruction errors for a fixed minibatch over the course of training: in each iteration (vertical slice) we plot the errors of each minibatch element. Thus, we can observe the distribution of errors. Orange points in the affine plots denote `inf` and `nan` values. $c$ denotes the condition number of the Jacobian, $\sigma_{max}$ and $\sigma_{min}$ denote the maximum and minimum singular values, respectively. At the top, we show the original images $x$ and in each snapshot, the reconstructions $\hat{x} = F^{-1}(F(x))$ and reconstruction errors $R = |x - \hat{x}|$. Note that the x- and y-axes differ for different settings, as the models become unstable at different points during training.

## 5 RELATED WORK

**Invertibility and stability of deep networks.** The inversion from activations in standard neural networks to inputs has been studied in various works, e.g. via optimization in input space (Mahendran & Vedaldi, 2014). Linking invertibility and inverse stability for relu-networks was e.g. done in Behrmann et al. (2018). However, few works study the stability of INNs: Gomez et al. (2017) study the numerical errors in the gradient computation when using their memory-efficient backpropagation variant. Similarly to our empirical analysis, (Jacobsen et al., 2018) computed the SVD of the Jacobian of a trained i-RevNet and observed an ill-conditioned Jacobian. Lastly, the i-ResNet architecture (Behrmann et al., 2019) yields bi-Lipschitz bounds by design.

On the other hand, the stability of neural networks has been of major interest due to the problem of exploding and vanishing gradients, and more recently due to adversarial examples (Szegedy et al., 2013) and training of Wasserstein GANs (Arjovsky et al., 2017). See e.g. (Anil et al., 2019) for a promising approach to learn flexible Lipschitz neural networks.

**Invertible building blocks.** Besides the invertible building blocks we studied in Table 1, several other approaches were proposed. Most prominently, autogressive models like MAF (Papamakarios et al., 2017) or IAF (Kingma et al., 2016) provide invertible models that are not studied in our analysis. Furthermore, several newer coupling layers that require numerical inversion have been introduced (Jaini et al., 2019; Durkan et al., 2019). Besides the coupling-based approaches, multiple approaches (Chen et al., 2018; Behrmann et al., 2019; Chen et al., 2019; Song et al., 2019) use numerical inversion schemes, where the interplay of numerical errors due to stability and errors due to the numerical approximation of the inverse adds another dimension to the study of invertibility.

**Fixed-Point arithmetic and limited precision.** Maclaurin et al. (2015); MacKay et al. (2018) implement invertible computation using fixed-point numbers, with specially-designed schemes to store information that is "lost" when bits are shifted due to multiplication/division, enabling exact invertibility at the cost of additional memory usage. As Gomez et al. (2017) point out, this approach allows exact numerical inversion when using additive coupling blocks independent of stability issues. However, our stability analysis aims for a broadly applicable methodology beyond the special case of additive coupling. Lastly, there may be connections to deep learning using limited precision, see e.g. (Gupta et al., 2015), which could provide more insights into our observed numerical errors.

## 6 CONCLUSION

Numerical instability is an important concern for the practical application of invertible models. If for instance analytical invertibility does not carry through to the numerical computation due to instabilities or numerical errors, the consequences can be arbitrarily severe. As shown in our experiments, this can impact memory-efficient backpropagation (Gomez et al., 2017) and thus significantly reduces the usability of invertible networks if not handled appropriately. Flow-GAN illustrates another application where non-invertibility poses a serious threat, as instabilities can strongly influence or even break likelihood-computation.

In this paper, we shed light on the underlying causes of instability by deriving Lipschitz bounds on many of the atomic building blocks commonly used to construct INNs. From a practical standpoint, we used diagnostics to measure stability and provided an empirical framework to benchmark stability. Further, we have shown how to guarantee stability for one of the most common INN architectures. We hope that this will inspire future work to view numerical stability as a crucial axis in the design of new building blocks and architectures for invertible neural networks.

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

# A  DERIVATIONS OF LIPSCHITZ BOUNDS

The bounds for invertible ResNets are taken from (Behrmann et al., 2019). For Neural ODEs (Chen et al., 2018), one needs to consider a Lipschitz constant $\text{Lip}(F)$ that holds for all $t \in [0, T]$, i.e.

$$\|F(t, x_1) - F(t, x_2)\|_2 \leq \text{Lip}(F)\|x_1 - x_2\|_2, \quad \text{for all} \quad t \in [0, T].$$

Then, the claimed bound is a standard result, see e.g. (Ascher, 2008, Theorem 2.3). Note that the inverse is given by $\frac{dy(t)}{dt} = -F(y(t), t)$, hence the same bound holds.

In the subsequent subsections, we derive the bounds for coupling layers.

## A.1  DERIVATION OF LIPSCHITZ BOUND FOR ADDITIVE COUPLING LAYERS

Consider an additive coupling block defined as

$$\begin{aligned}
F(x)_{I_1} &= x_{I_1} \\
F(x)_{I_2} &= x_{I_2} + g(x_{I_1}),
\end{aligned}$$

where $I_1, I_2$ is a disjoint partition of indices $\{1, ..., d\}$ of the same cardinality, i.e. $|I_1| = |I_2| = \frac{d}{2}$. Further, $x_{I_1}, x_{I_2}$ correpsonds to the corresponding dimension of $x \in \mathbb{R}^d$ and $g : \mathbb{R}^{\frac{d}{2}} \to \mathbb{R}^{\frac{d}{2}}$. By Lemma 3, it is

$$\text{Lip}(F) = \sup_{x \in \mathbb{R}^d} \|J_F(x)\|_2.$$

Thus, in order to obtain a bound on the Lipschitz constant, it is helpful to look into the structure of the Jacobian. If the partitions $I_1$ and $I_2$ correspond to the first and last $\frac{d}{2}$ indices, the Jacobian has a lower-block structure with an identity diagonal, i.e.

$$J_F(x) = \begin{pmatrix} I & 0 \\ J_g(x) & I \end{pmatrix}.$$

By using this structure, we can derive the following upper bound:

$$\begin{aligned}
\text{Lip}(F)^2 &= \sup_{x \in \mathbb{R}^d} \|J_F(x)\|_2^2 \\
&= \sup_{x \in \mathbb{R}^d} \sup_{\|x^*\|_2 = 1} \|J_F(x)x^*\|_2^2 \\
&= \sup_{x \in \mathbb{R}^d} \sup_{\|x^*\|_2 = 1} \|(J_F(x)x^*)_{I_1}\|_2^2 + \|(J_F(x)x^*)_{I_2}\|_2^2 \\
&= \sup_{x \in \mathbb{R}^d} \sup_{\|x^*\|_2 = 1} \|x_{I_1}^*\|_2^2 + \|x_{I_2}^* + J_g(x)x_{I_1}^*\|_2^2 \\
&\leq \sup_{x \in \mathbb{R}^d} \sup_{\|x^*\|_2 = 1} \|x_{I_1}^*\|_2^2 + (\|x_{I_2}^*\|_2 + \|J_g(x)x_{I_1}^*\|_2)^2 & (6) \\
&= \sup_{x \in \mathbb{R}^d} \sup_{\|x^*\|_2 = 1} \|x_{I_1}^*\|_2^2 + \|x_{I_2}^*\|_2^2 + 2\|x_{I_2}^*\|_2\|J_g(x)x_{I_1}^*\|_2 + \|J_g(x)x_{I_1}^*\|_2^2 \\
&= \sup_{x \in \mathbb{R}^d} \sup_{\|x^*\|_2 = 1} \|x^*\|_2^2 + 2\|x_{I_2}^*\|_2\|J_g(x)x_{I_1}^*\|_2 + \|J_g(x)x_{I_1}^*\|_2^2 \\
&= \sup_{x \in \mathbb{R}^d} \sup_{\|x^*\|_2 = 1} 1 + 2\|x_{I_2}^*\|_2\|J_g(x)x_{I_1}^*\|_2 + \|J_g(x)x_{I_1}^*\|_2^2 \\
&= \sup_{x \in \mathbb{R}^d} \sup_{\|x^*\|_2 = 1} 1 + 2\|J_g(x)x_{I_1}^*\|_2 + \|J_g(x)x_{I_1}^*\|_2^2 \\
&= \sup_{x \in \mathbb{R}^d} \sup_{\|x^*\|_2 = 1} \left(1 + \|J_g(x)x_{I_1}^*\|_2\right)^2 \\
&= \sup_{x \in \mathbb{R}^d} \left(1 + \|J_g(x)\|_2\right)^2 \\
\Rightarrow \text{Lip}(F) &\leq 1 + \text{Lip}(g).
\end{aligned}$$

Furthermore, the inverse of $F$ can be obtained via the simple algebraic transformation ($y := F(x)$)

$$F^{-1}(y)_{I_1} = y_{I_1}$$
$$F^{-1}(y)_{I_2} = y_{I_2} - g(y_{I_1}).$$

Since the only difference to the forward mapping is the minus sign, the Lipschitz bound for the inverse is the same as for the forward mapping.

## A.2 DERIVATION OF LIPSCHITZ BOUND FOR AFFINE COUPLING LAYERS

Since the structure of the forward and inverse mapping for affine coupling layers has some differences, we split the derivation of the Lipschitz bounds into two sections. First, we start with the forward mapping and then reuse several steps for the bounds on the inverse mapping.

### A.2.1 DERIVATION FOR THE FORWARD MAPPING

Consider an affine coupling block defined as

$$F(x)_{I_1} = x_{I_1}$$
$$F(x)_{I_2} = x_{I_2} \odot g(s(x_{I_1})) + t(x_{I_1}),$$

where $g(\cdot) \neq 0$ for all $X_{I_2}$ and $I_1, I_2$ as before. The Jacobian for this operation has the structure

$$J_F(x) = \begin{pmatrix} I & 0 \\ D_I(x_{I_2})D_{g'}(x_{I_1})J_s(x_{I_1}) + J_t(x_{I_1}) & D_g(s(x_{I_1})) \end{pmatrix},$$

where $D$ are following diagonal matrices

$$D_I(x_{I_2}) = \text{diag}\left((x_{I_2})_1, \ldots, (x_{I_2})_{|I_2|}\right),$$
$$D_{g'}(x_{I_1}) = \text{diag}\left(g'(s(x_{I_2})_1, \ldots, g'(s(x_{I_2})_{|I_2|})\right)$$
$$D_g(s(x_{I_1})) = \text{diag}\left(g(s(x_{I_2})_1, \ldots, g(s(x_{I_2})_{|I_2|})\right).$$

Denote

$$M(x) := D_I(x_{I_2})D_{g'}(x_{I_1})J_s(x_{I_1}) + J_t(x_{I_1}).$$

By using an analogous derivation as in equation 6 (up to the inequality sign), we get

$$\begin{aligned}
\text{Lip}(F)^2 &\leq \sup_{x \in \mathbb{R}^d} \sup_{\|x^*\|_2=1} \|x_{I_1}^*\|_2^2 + \left(\|D_g(s(x_{I_1}))x_{I_2}^*\|_2 + \|M(x)x_{I_1}^*\|_2\right)^2 \\
&= \sup_{x \in \mathbb{R}^d} \max_{i \in [|I_1|]} (1, D_g(s(x_{I_1})_i))^2 + 2 \max_{i \in [|I_1|]} (D_g(s(x_{I_1})_i))\|M(x)\|_2 + \|M(x)\|_2^2 \\
&\leq \sup_{x \in \mathbb{R}^d} \max_{i \in [|I_1|]} (1, D_g(s(x_{I_1})_i))^2 + 2 \max_{i \in [|I_1|]} (1, D_g(s(x_{I_1})_i))\|M(x)\|_2 + \|M(x)\|_2^2 \\
&= \sup_{x \in \mathbb{R}^d} \left(\max_{i \in [|I_1|]} (1, D_g(s(x_{I_1})_i)) + \|M(x)\|_2\right)^2 \\
\iff \text{Lip}(F) &\leq \max_{i \in [|I_1|]} (1, D_g(s(x_{I_1})_i)) + \sup_{x \in \mathbb{R}^d} \|M(x)\|_2.
\end{aligned}$$

Next, we will look into the structure of $M(x)$ to derive a more precise bound. Since inputs $x$ are assumed to be bounded as $x \in [a, b]^d$, it holds

$$\|D_I(x_{I_2})\|_2 \leq \max(|a|, |b|).$$

Furthermore, let the derivative $g'$ of the element-wise function $g$ be globally bounded by $c$, i.e. $\sup_{x \in \mathbb{R}} g'(x) \leq c_{g'}$. Then, it is

$$\|D_{g'}(x_{I_1})\|_2 \leq c_{g'}.$$

In a similar manner as in section A.1, the spectral norm of the Jacobian of the scale-function $s$ and translation-function $t$ can be bounded by their Lipschitz constant, i.e.

$$\|J_s(x_{I_1})\|_2 \leq \mathrm{Lip}(s)$$
$$\|J_t(x_{I_1})\|_2 \leq \mathrm{Lip}(t).$$

By using above bounds, we obtain

$$\sup_{x \in \mathbb{R}^d} \|M(x)\|_2^2 \leq \max(|a|, |b|) \cdot c \cdot \mathrm{Lip}(s) + \mathrm{Lip}(t).$$

If we further assume, that the elementwise-function $g$ is globally upper bounded by $c_g$ and we insert above bounds, we obtain

$$\mathrm{Lip}(F) \leq \max(1, c_g) + \max(|a|, |b|) \cdot c_{g'} \cdot \mathrm{Lip}(s) + \mathrm{Lip}(t).$$

### A.2.2  DERIVATION FOR THE INVERSE MAPPING

For the affine coupling block from section A.2.1, the inverse is defined as

$$F^{-1}(y)_{I_1} = y_{I_1}$$
$$F^{-1}(y)_{I_2} = (y_{I_2} - t(x_{I_1})) \oslash g(s(y_{I_1})),$$

where $g(\cdot) \neq 0$ for all $X_{I_2}, I_1, I_2$ as before and $\oslash$ denotes elementwise division. The Jacobian for this operation has the structure

$$J_F(x) = \begin{pmatrix} I & 0 \\ M^*(y) & D_{\frac{1}{g}}(s(x_{I_1})) \end{pmatrix},$$

where $D_{\frac{1}{g}}(s(x_{I_1}))$ denotes a diagonal matrix, as before. Furthermore, $M*$ is defined as

$$M^*(y) = D_I(y_{I_2}) D_{(\frac{1}{g})'}(s(y_{I_1})) J_s(y_{I_1}) - D_{(\frac{1}{g})'}(s(y_{I_1})) J_s(y_{I_1}) D_I(t(y_{I_1})) - D_{\frac{1}{g}}(s(y_{I_1})) J_t(y_{I_1}),$$

where $D_{(\frac{1}{g})'}(s(x_{I_1}))$ also denotes a diagonal matrix. Using analogous arguments as in section A.2.1, we obtain the bound

$$\mathrm{Lip}(F^{-1}) \leq \max_{i \in [|I_1|]} (1, D_{\frac{1}{g}}(s(x_{I_1})_i)) + \sup_{x \in \mathbb{R}^d} \|M^*(x)\|_2.$$

Hence, we need to further bound the spectral norm of $M^*$. First, assume that $\frac{1}{g}$, the derivative $\left(\frac{1}{g}\right)'$ and translation $t$ is globally upper bounded by $c_{\frac{1}{g}}$, $c_{(\frac{1}{g})'}$ and $c_t$ respectively. Furthermore consider bounded inputs $y \in [a^*, b^*]^d$. Then we obtain the bound

$$\sup_{x \in \mathbb{R}^d} \|M^*(x)\|_2^2 \leq \max(|a^*|, |b^*|) \cdot c_{(\frac{1}{g})'} \cdot \mathrm{Lip}(s) + c_{(\frac{1}{g})'} \cdot \mathrm{Lip}(s) \cdot c_t + c_{\frac{1}{g}} \cdot \mathrm{Lip}(t).$$

Hence, we can bound the Lipschitz constant of the inverse of an affine block as

$$\mathrm{Lip}(F^{-1}) \leq \max_{i \in [|I_1|]} (1, c_{\frac{1}{g}}) + \max(|a^*|, |b^*|) \cdot c_{(\frac{1}{g})'} \cdot \mathrm{Lip}(s) + c_{(\frac{1}{g})'} \cdot \mathrm{Lip}(s) \cdot c_t + c_{\frac{1}{g}} \cdot \mathrm{Lip}(t).$$

## B  NUMERICAL ERRORS AND LIPSCHITZ CONSTANTS

In a general setting, connecting numerical errors e.g. due to floating point operations to Lipschitz constants of the underlying mapping in a quantitative manner is not straightforward. For example, numerical errors due to limited precision occurs when summing to floating point numbers. As discussed in (Gomez et al., 2017), this occurs in additive coupling layers and is one source of numerical errors we observe in our experiments.

To formalize the connection to the Lipschitz constant, consider the following two mappings:

$$F(x) = z, \quad \text{(analytical exact computation)}$$
$$F_\delta(x) = z + \delta =: z_\delta, \quad \text{(floating point inexact computation)}$$

In order to bound the error in the reconstruction due to the imprecision in the forward mapping, let $x_{\delta_1} = F^{-1}(z_\delta)$. Now consider

$$\|x - x_{\delta_1}\|_2 \leq \mathrm{Lip}(F^{-1})\|z - z_\delta\|_2 = \mathrm{Lip}(F^{-1})\|\delta\|_2,$$

where the Lipschitz constant of the inverse is used to bound the influence of the numerical error in the forward mapping. However, similarly to the forward mapping, the inverse mapping can also be imprecise. Thus, we introduce

$$F_\delta^{-1}(z_\delta) = x_{\delta_1} + \delta_2 := x_{\delta_2}$$

to formalize the numerical error in the inverse mapping. Hence, we obtain the bound

$$
\begin{aligned}
\|x - (x_{\delta_1} + \delta_2)\|_2 &\leq \|x - x_{\delta_1}\|_2 + \|\delta_2\|_2 \\
&\leq \mathrm{Lip}(F^{-1})\|z - z_\delta\|_2 + \|\delta_2\|_2 \\
&= \mathrm{Lip}(F^{-1})\|\delta\|_2 + \|\delta_2\|_2,
\end{aligned}
$$

where the numerical errors of the mapping are denoted via $\delta$ (forward) and $\delta_2$ (inverse). While obtaining quantitative values for $\delta$ and $\delta_2$ for a model as complex as deep neural networks is hard, above formalization still provides insights into a potential role of the inverse stability when reconstructing inputs.

## C  ADDITIONAL DETAILS FOR CLASSIFICATION EXPERIMENTS

Both affine and additive coupling-based models have the same architecture, that consists of 3 levels, 16 blocks per level, and 128 hidden channels. Each level consists of a sequence of residual blocks that operate on the same dimensionality. Between levels, the input is spatially downsampled by $2\times$ in both width and height, while the number of channels is increased by $4\times$. Each residual block consists of a chain of $3 \times 3$, $1 \times 1$, $3 \times 3$ convolutions, with ReLU activations in between. We trained on CIFAR-10 for 200 epochs, using SGD with Nesterov momentum 0.9 and weight decay 5e-4, with initial learning rate 0.01, decayed by a factor of 5 at epochs 60, 120, and 160. We found that

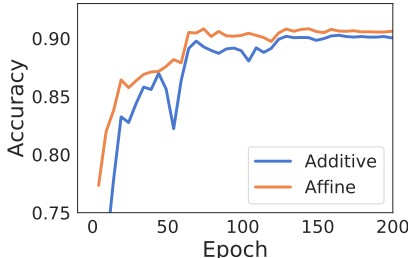

Figure 6: **Test classification accuracy on CIFAR-10.**

using a smaller initial learning rate of 0.01 was important for training the INN classifier, as opposed to the standard initial learning rate of 0.1 used for ResNets (Zagoruyko & Komodakis, 2016). We used standard data augmentation (random cropping and horizontal flipping).

## D  EXPERIMENTAL DETAILS FOR FLOW-GAN

For the experiments in Section 4.3, the ADV models are trained with standard binary cross entropy loss and the hyperparameters in Table 4, whereas the MLE models are trained with learning rate 1e-3.

| Variable | Values |
|---|---|
| batch size | 64 |
| learning rate (generator) | 1e-5 |
| learning rate (discriminator) | 1e-4 |
| weight decay (both) | 0 |
| optimizer | Adam(0.5, 0.99) |

Table 4: Hyperparameters for ADV models

# E    EXTENDED RESULTS FOR CRAFTED, NON-INVERTIBLE INPUTS

**PGD Setup.**    To find non-invertible inputs for Glow and Residual Flows, we used PGD (Eq. 3) with $\epsilon = 0.1$ and step size 0.01. For the Glow model in Section 4.2, we consistently found inputs with severe reconstruction errors (as shown in Figure 3) in fewer than 10 PGD iterations. For the Residual Flow model analyzed in this section, we ran 200 iterations of PGD. In each iteration, the pixel values of the perturbed image were clipped to the valid input range that the respective model was trained on ($[-0.5, 0.5]$ for Glow and $[0, 1]$ for the Residual Flow).

**Crafted Inputs for Residual Flows.**    We also applied the PGD attack from Section 4.2 to a Residual Flow (Chen et al., 2019) pre-trained on CIFAR-10 (Figure 7).[4] We find that, while there are visible differences between the crafted input $x'$ and its reconstruction $\hat{x}'$, the reconstruction errors are less severe for the Residual Flow compared to Glow (analyzed in Section 4.2).

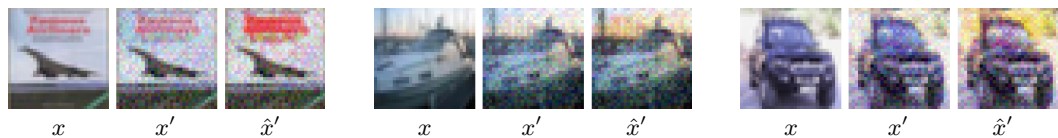

$x \qquad x' \qquad \hat{x}' \qquad\qquad x \qquad x' \qquad \hat{x}' \qquad\qquad x \qquad x' \qquad \hat{x}'$

Figure 7: **Crafting non-invertible inputs for a CIFAR-10 Residual Flow model.**

**Instability Outside the Range of Training Inputs.**    Here, we applied the PGD attack from Section 4.2 to a Glow model pre-trained on Celeb-A (Liu et al., 2015). [5] This model was trained on images normalized to the range $[-0.5, 0.5]$. While the PGD attack was not successful at finding adversarial inputs in $[-0.5, 0.5]$, it succeeded when the range was increased to $[-0.7, 0.7]$ (by using $\epsilon = 0.2$ and not clipping the perturbed inputs to $[-0.5, 0.5]$), yielding the example shown in Figure 8. Thus, we found that invertible models can become numerically non-invertible on out-of-distribution data.

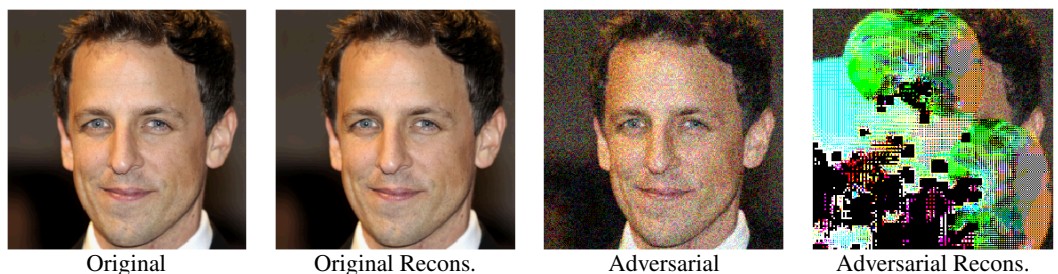

Original        Original Recons.        Adversarial        Adversarial Recons.

Figure 8: **Glow becomes non-invertible for inputs outside the training distribution.** We use a SOTA Glow model trained on images normalized to $[-0.5, 0.5]$, and use PGD to find an adversarial input constrained to the *larger range* $[-0.7, 0.7]$. This attack succeeds in finding examples that induce dramatic reconstruction error.

---

[4]We used the pre-trained model from `https://github.com/rtqichen/residual-flows`.
[5]We used the model from `https://github.com/openai/glow`.

# F    Motivating the Decorrelation Task

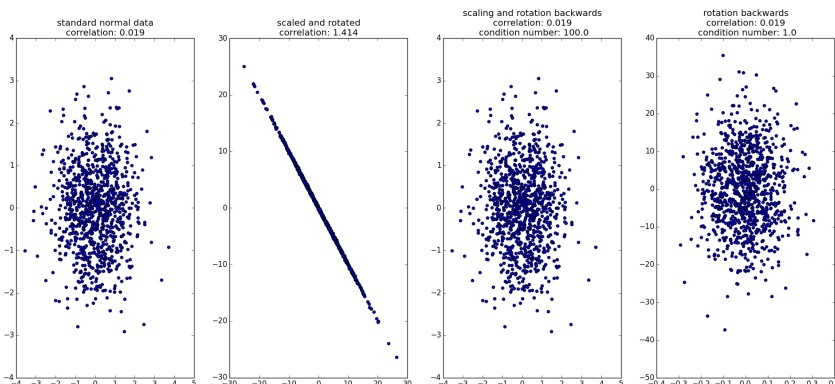

Figure 9: Toy example to motivate the decorrelation task. **Left:** standard normal data, **2. left:** scaled by $D$ and rotated data by $R$, **2. right:** scaling and rotation backwards, low correlation but higher condition number, **right:** rotation backwards, low correlation and low condition number.

To motivate the decorrelation task as simple toy environment for stability of invertible models consider the following task (and its visualization in Figure 9):

- Consider input data $x$ that is distributed via a standard normal distribution, i.e. $x \sim \mathcal{N}(0, I)$ (Figure 9 (left)).
- Assume the data is transformed by a rotation matrix $R$ and a diagonal matrix $D$, i.e. $y = RDx$ (Figure 9 (2. from left)).
- Goal: decorrelate transformed data $y$ using an invertible mapping.

Since correlation is independent of scale (scaling by standard deviation of the data), at least the two solutions $A_2 = D^{-1}R^T$ (Figure 9 (2. from right)) and $A_1 = R^T$ (Figure 9 (right)) are equally valid for the given decorrelation task. However, the conditioning of the mappings $A_1$ and $A_2$ can be largely different if the scaling matrix $D$ has a high condition number. Hence, this task both offers a stable solution, namely $A_1$, and a (potentially) unstable solution $A_2$.

To conclude, decorrelation can allow multiple solutions with different stability. Hence, decorrelation is a natural simple task to study which solution the INN picks. Furthermore, guiding the network to a stable solution is a justified strategy for this task and it is not expected to harm performance.

# G    Decorrelation Example Code

Here we provide an example implementation of the decorrelation objective used in Section 4.4, that minimizes the norm of the off-diagonal entries in the correlation matrix.

Listing 1: Example PyTorch code to implement the decorrelation loss used in our experiments.

```
z = model(img)
z_flat = z.view(z.size(0), -1)
z_flat = z_flat - z_flat.mean(dim=0) # Subtract mean
z_flat = z_flat / (z_flat.std(dim=0) + 1e-8) # Standardize
correlation = (torch.mm(z_flat.t(), z_flat) / (z_flat.size(0)-1))
loss = torch.norm(correlation - torch.diag(torch.diagonal(correlation)))
optimizer.zero_grad()
loss.backward()
optimizer.step()
```

## H    EXTENDED RESULTS FOR DECORRELATION

**Decorrelation Experiment Details**    Here we provide additional details on the model architectures and training schemes we used in our numerical experiments.

For all the decorrelation experiments, we used a 3-level model with blocks of depth 16. We used Adam (Kingma & Ba, 2015) with fixed learning rate 1e-4 and no weight decay, and trained on mini-batches of size 64. The CIFAR-10 images were normalized to the range [-0.5, 0.5], and were dequantized with uniform noise in [0, 1e-6].

**Effect of Model Depth.**    Furthermore, we investigated the effect of network depth on stability since its an additional influence factor besides the selection of each invertible building block. Starting with a 3-level additive model with ActNorm, we vary the depth of the blocks between $\{4, 16, 32\}$ and train with the decorrelation objective. The quantitative reconstruction errors and condition numbers of the Jacobians are shown in Figure 10. As expected, deeper architectures become unstable faster than shallow ones.

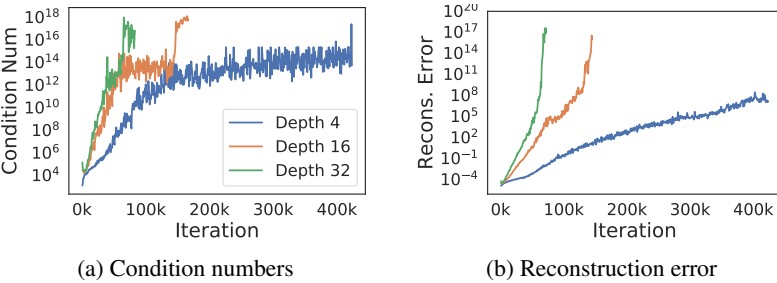

(a) Condition numbers          (b) Reconstruction error

Figure 10: **Comparing stability of additive flows of different depths.** These models all have ActNorm both between and inside the additive blocks.

**Loss Plots.**    Here we show that all the model variants investigated in the decorrelation experiments achieve their objective, i.e., the loss decreased enough for the correlation matrices to be diagonal.

**Evolution of Condition Numbers, Max & Min Singular Values for Decorrelation.**    Here we plot the condition numbers, maximum and minimum singular values during training for the decorrelation task. We include plots for all settings discussed in the main paper as well as the appendix.

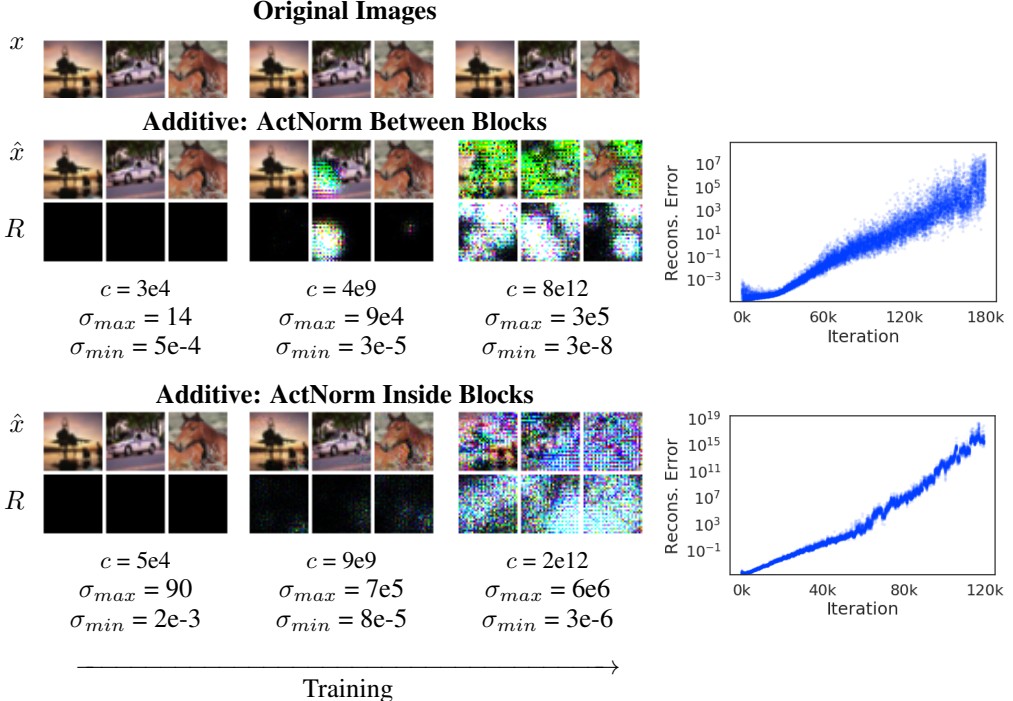

Figure 11: Additional settings for additive coupling, where: 1) ActNorm was applied only between blocks; and 2) ActNorm was applied only inside blocks. Both models become unstable and exhibit severe reconstruction artifacts.

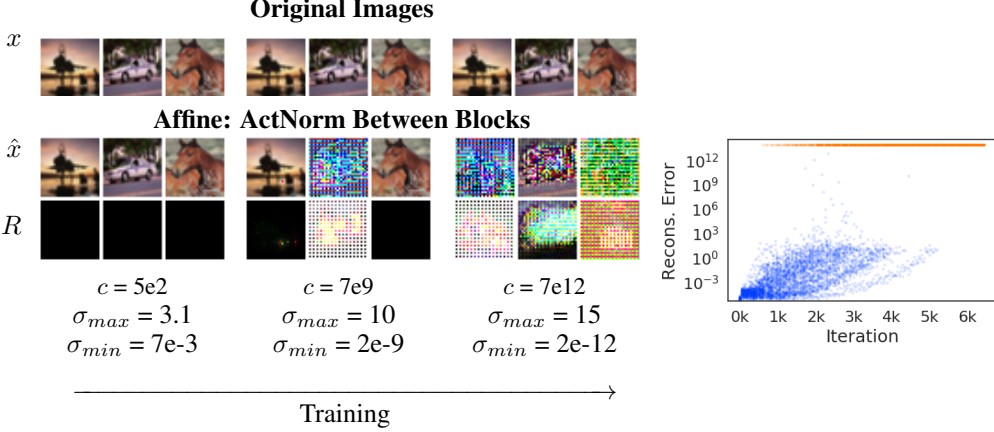

Figure 12: Additional setting for affine coupling, where ActNorm was applied only between blocks. Similarly to the other affine settings described in Section 4.4, this model becomes highly unstable rapidly during training.

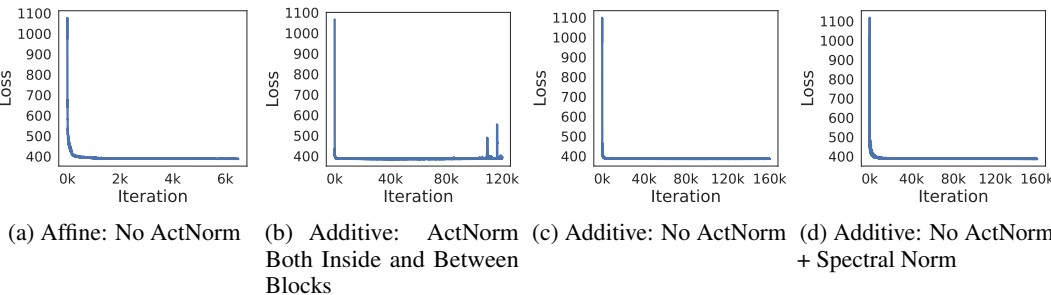

(a) Affine: No ActNorm (b) Additive: ActNorm Both Inside and Between Blocks (c) Additive: No ActNorm (d) Additive: No ActNorm + Spectral Norm

Figure 13: **Decorrelation loss plots.** These plots track the norm of the off-diagonal entries in the correlation matrix while training with the decorrelation objective.

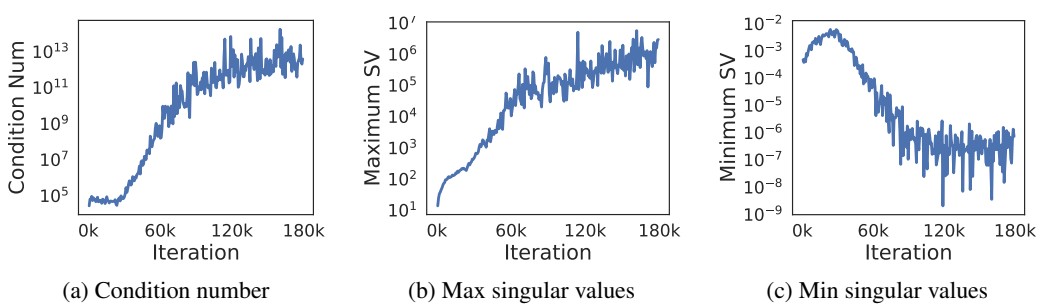

(a) Condition number (b) Max singular values (c) Min singular values

Figure 14: **Additive: ActNorm Between Blocks**

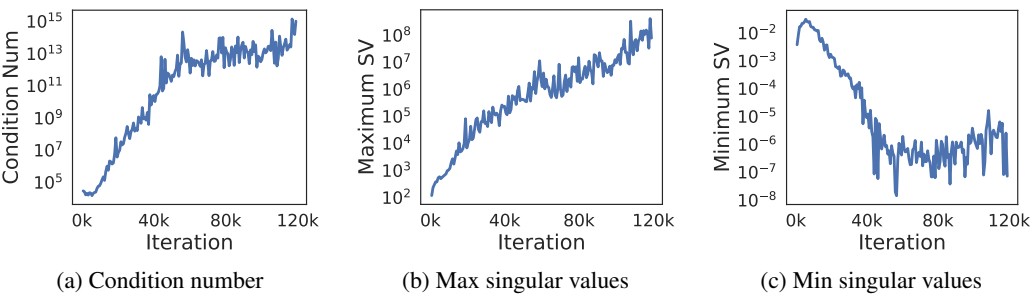

(a) Condition number (b) Max singular values (c) Min singular values

Figure 15: **Additive: ActNorm Both Inside and Between Blocks**

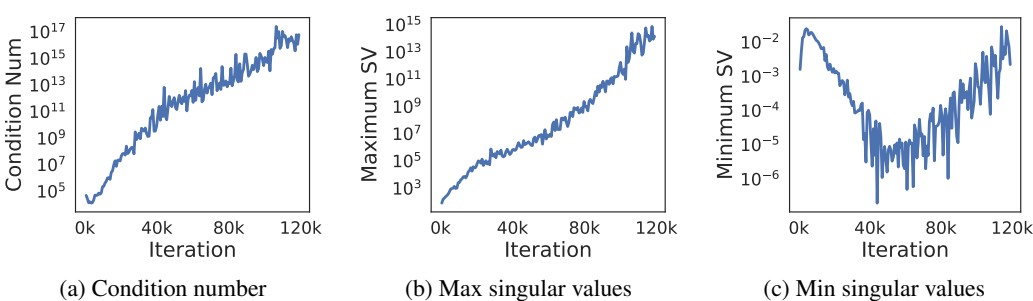

(a) Condition number (b) Max singular values (c) Min singular values

Figure 16: **Additive: ActNorm Inside Blocks**

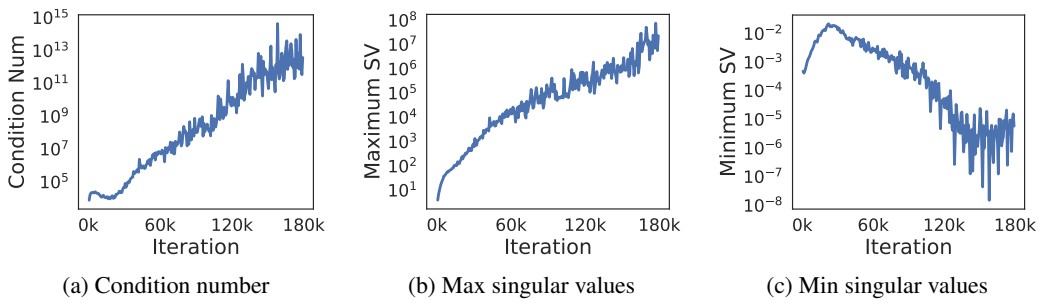

(a) Condition number          (b) Max singular values          (c) Min singular values

Figure 17: **Additive: No ActNorm**

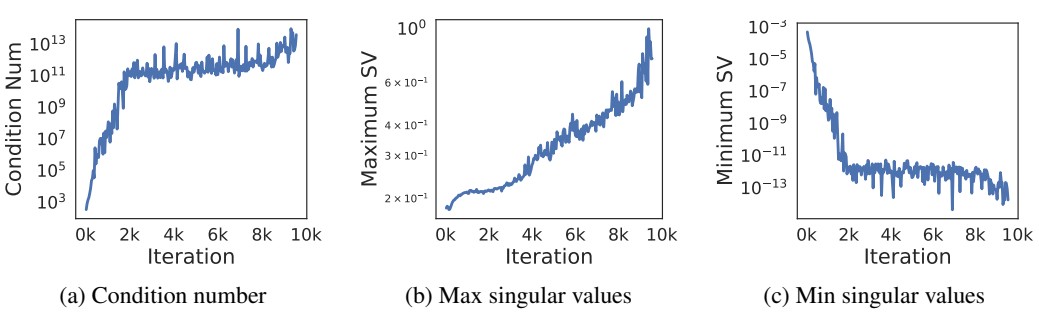

(a) Condition number          (b) Max singular values          (c) Min singular values

Figure 18: **Affine: No ActNorm**

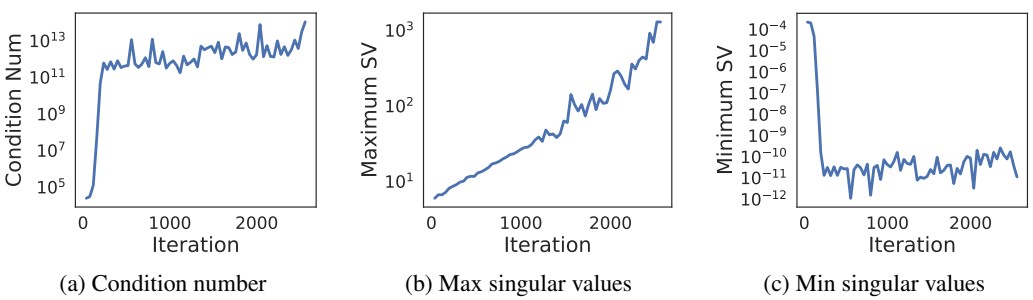

(a) Condition number          (b) Max singular values          (c) Min singular values

Figure 19: **Affine: ActNorm Both Inside and Between Blocks**

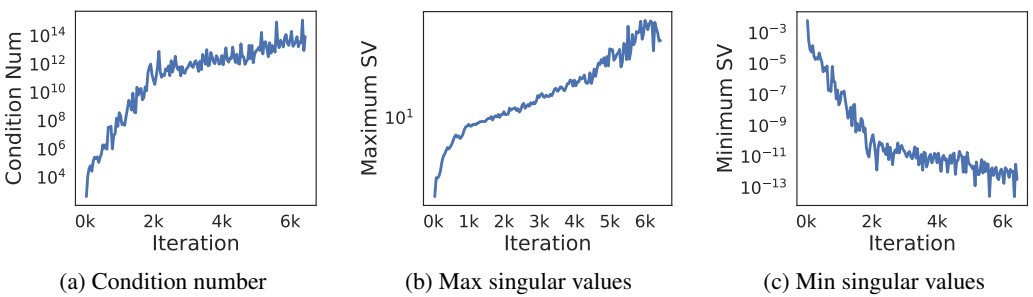

(a) Condition number          (b) Max singular values          (c) Min singular values

Figure 20: **Affine: ActNorm Between Blocks**

## I    REFITTING PRIOR IN FLOW-GAN

In general, if the model is not optimized with forward KL, $D_{KL}(P_{\text{data}}||P_\theta)$, as in the case when optimizing with maximum likelihood, we cannot be sure $F(x)$,    $x \sim P_{\text{data}}$ is best fitted with a standard Normal. Hence, a reasonable strategy, without changing anything in the learned network, is to refit the prior parameters. Here we simply optimize for maximum likelihood (as typically done for flow models) while only fitting a diagonal variance in prior. This can be interpreted as increasing the entropy of our model. In Kingma & Dhariwal (2018), they observed the opposite phenomenon that a model trained with maximum likelihood generates better samples after decreasing the entropy in the prior. See Figure 21 for samples after refitting the prior.

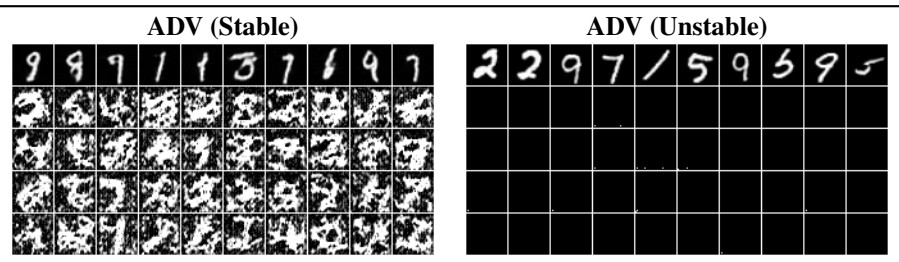

Figure 21: The row number corresponds to the number of epochs after refitting the prior variance. The **unstable** model fails to generate samples (i.e., outside of valid pixel values) after the prior is refitted, whereas even though the **stable** model degrades in sample quality, it is able to generate valid images.

