# OpenReview forum: "On the Invertibility of Invertible Neural Networks"
_ICLR.cc/2020/Conference — Reject_

### Official Review · AnonReviewer3 · 2019-10-22
**Official Blind Review #3**

**Rating:** 3

**Review:**

The paper claims that for invertible neural networks, mathematical guarantees on invertibility is not enough, and we also require numerical invertibility. To this end, the lipschitz constants/condition numbers of Jacobians of both the forward and inverse maps of invertible NNs based on coupling layers are examined mathematically and experimentally. The paper also displays cases that expose non-invertibility in these architectures via gradient-based construction of adversarial inputs, as well as a decorrelation benchmark task, and show that spectral normalization can be a remedy for stabilizing these flows.

I think it’s a good point that we need to monitor the Lipschitz constant/bounds of both directions of these invertible functions. It’s true that the focus for stabilising NNs by bounding Lipschitz constants was always on the forward function, and for invertible functions we should also ensure that the inverse is numerically stable to compute.

The mathematical contribution of the paper is twofold - 1. deriving bounds on the lipschitz constants of the forward and inverse mapping of additive/affine coupling blocks 2. summarising known lipschitz bounds of forward and inverse mappings of other invertible layers (iResNet, neuralODE, invertible 1x1 convolutions etc). The main contribution lies in 1, and the derivation for the additive coupling block (volume preserving) is neat (although fairly straightforward), but the derivation for the affine coupling layer (NVP) is not useful nor insightful; they are local Lipschitz bounds (so require bounds on all intermediate activations, which is difficult as pointed out by the authors), and the numerical value of this bound was not used at all in relation to the numerical experiments - I imagine the bound is loose. Given that it seems difficult to find a tight global lipschitz bound, I think it would be more insightful to compute a lower bound to the lipschitz constant of the model (with fixed parameter values) by maximising the spectral norm of the Jacobian with respect to the inputs (or outputs if looking at the inverse map) - this will yield a lower bound by Lemma 3. This will be numerical, but more informative since it will give you an indication of where in the input space (or output space if looking at the inverse) there could be numerical instabilities. Also I think the bound on the local lipschitz constant of the inverse for the affine coupling block might be incorrect, because in A.1.1, the inverse map is F^{-1}(y)_I1 = y_I1, F^{-1}(y)_I2 = (y_I2 - t(y_I1))/g(s(y_I1)), so the scale and shift is s’(y_I1) := 1/g(s(y_I1)) and t’(y_I1):=- t(y_I1)/g(s(y_I1)), and hence I think this needs to be taken into account for computing the lipschitz bound of the inverse

I have mixed feelings about the experimental section. In section 4.1, it is interesting to see that we can find inputs where trained flow models can show numerical non-invertibility, evident in the poor reconstructions. It would be a nice addition to investigate whether this is coming from the forward function or its inverse, by examining the norm of the Jacobian of F and F^{-1} at the input x_delta and output F(x_delta) respectively.

However, the decorrelation task introduced in section 4.2 is puzzling. I don’t understand why for these invertible models, you are investigating invertibility for parameter values trained to decorrelate, as opposed to parameter values used in the usual task of density estimation with flows (or any other standard application of invertible NNs). The two reasons given in the paper are that 1) decorrelation is a simpler task and 2) it allows both stable and unstable transforms as solutions, but these are not convincing. Point 2) holds for flow-based density estimation as well, and regarding point 1), density estimation is the task we usually care about when using invertible NNs, and this is also computationally plausible/tractable, whereas even if decorrelation is a simpler task, it’s not a task that users of invertible NNs are interested in. It is good to know that these invertible NN architectures CAN admit values that are numerically non-invertible, but I would be much more interested to know whether this actually holds when they have been trained for flow-based density estimation. I’m not sure whether the experimental results on models trained for the decorrelation task are useful, because a model that is stable when trained for the decorrelation task may be unstable when trained for flow-based density estimation and vice versa. The observation that spectral normalization can help address numerical instability is useful, but from the perspective of someone who wants to use these invertible NNs for density estimation, I would like to know what is the sacrifice in expressivity/validation performance (if any) when using spectral normalization in these invertible architectures. Also, the results would be more relevant if the architectures resembled the architectures used for invertible models used in the literature (e.g. GLOW) where we not only have coupling layers but they are interleaved with PLU linear flows.

In section 5, the result that Flow-GANs can be numerically non-invertible is more relevant, and it is useful to know that spectral normalisation can help resolve this issue, but again it would be useful to quantify whether this comes at the cost of the quality of generated samples (Figure 3 shows several samples, but a more thorough quantitative & qualitative comparison would be welcome). Also regarding the point about likelihood in Section 5, where the authors state “it cannot be trusted as true likelihood due to lack of invertibility”, I think it should be emphasised that this point holds specifically for flow-GANs where for F: z -> x, you need a numerically accurate F^{-1} to compute the density, but for standard flow-based density estimation where F:x -> z, you never need to compute the inverse for computing the likelihood, hence if F has a small lipschitz constant then the likelihood will be accurate, regardless of whether the inverse is numerically stable or not.

Overall I believe the experimental section can be largely improved, and given that the motivation of the paper is nice and the paper is clearly written and nicely presented, it would be a shame to leave the experiment section as it is.

Minor typos/Qs:
p2: this problems <- this problem
p8: and with maximum likelihood (ML) - should this be removed?
p13: t(x_I2) <- t(x_I1)


**Experience Assessment:**

I have read many papers in this area.

**Review Assessment: Checking Correctness Of Derivations And Theory:**

I carefully checked the derivations and theory.

**Review Assessment: Checking Correctness Of Experiments:**

I carefully checked the experiments.

**Review Assessment: Thoroughness In Paper Reading:**

I read the paper thoroughly.

---

> ### Author Response · Authors · 2019-11-14
> **Response to R3 (comment 1)**
>
> Thank you for your insightful review. We appreciate your concerns and respond to them below:
>
> Q: Overall I believe the experimental section can be largely improved.
> ---------------------
> A: We restructured the experimental section based on four different tasks, please see our summary of the revision. The main change is the new classification section (Section 4.1), where we demonstrate that INN-based classifiers can become non-invertible on CIFAR-10. This in turn has severe impact when using the inverse for memory-efficient backpropagation as we show in Figure 1. Furthermore, we moved the decorrelation section to the end (Section 4.4) in order to benchmark different architecture settings as a final ablation study.
> ---------------------
>
> Q: Given that it seems difficult to find a tight global Lipschitz bound, I think it would be more insightful to compute a lower bound to the Lipschitz constant of the model (with fixed parameter values) by maximising the spectral norm of the Jacobian with respect to the inputs (or outputs if looking at the inverse map) - this will yield a lower bound by Lemma 3. This will be numerical, but more informative since it will give an indication of where in the input space (or output space if looking at the inverse) there could be numerical instabilities.
> ---------------------
> A: This is indeed a very interesting idea and is likely to give further insights into the local stability. Yet, we have to leave such a study to future work since we would first need to find a suitable approach to solve this optimization problem appropriately. For example, one could approximate the spectral norm of the Jacobian using power-iteration and then backpropagate through this operation. It is, however, unclear if this approach converges due to potential interactions of the two iterations (power iteration and gradient descent).
> ---------------------
>
> Q: I think the bound on the local lipschitz constant of the inverse for the affine coupling block might be incorrect
> ---------------------
> A: This is indeed an error, thank you for spotting it. We added a derivation to the Appendix A.2.2 and corrected the local Lipschitz bound in Table 1 (see our summary of revisions above).
> ---------------------
>
> Q: The derivation for the affine coupling layer (NVP) is not useful nor insightful.
> ---------------------
> A: Indeed, the derivation is quite technical and the provided bounds might be loose. However, we think that these bounds are a key result for multiple reasons:
>
> 1) They show how affine coupling is fundamentally different from standard neural networks with respect to stability. While almost all neural networks are globally Lipschitz continuous, this *does not* hold for affine coupling anymore. Only local bounds can be obtained.
> 2) Those local bounds may still provide guidance on how to stabilize affine blocks: in addition to bounding the Lipschitz norms of the mapping, one needs to bound the inputs to the layer.
> 3) They show a tradeoff between higher expressivity (nonlinear scaling in affine blocks) vs. instability (only local Lipchitz bounds).
> ---------------------
>
> Q: Investigate whether the non-invertibility of the crafted inputs comes from the forward or inverse mapping
> ---------------------
> A:  The forward mapping introduces some numerical imprecision z_\delta (but no nan or inf values). This imprecision is then amplified by the unstable inverse, which results in nan and inf values. Hence, the issues seem to occur both due to issues with the forward and especially with the inverse mapping. To better understand this effect we found the maximum singular value of the forward mapping to be 5488.1636 (locally for an input) and the smallest singular value was 1.18e-10, which confirms that the inverse is locally very unstable.
>
>
> Q: Why use decorrelation as opposed to density estimation?
> ---------------------
> A:  We do also have results related to density estimation. In Section 4.2 we analyze the invertibility of trained SOTA density models by crafting inputs. Furthermore, we included BPD and FID results for MLE-trained models in Section 4.3 (Table 2).
>
> However, we believe that decorrelation is a useful task to better understand the effects that influence the stability of INNs. While much energy has been invested in designing architectures and training settings for common tasks such as INN-based density estimation, some of these settings fail for other objectives, as we show with our decorrelation example. Furthermore, decorrelation is still related to density estimation, which is why we view this as an ablation study and a simple task to benchmark the influence of different architecture settings under a less standard objective.
>
> In our revised manuscript we have moved the decorrelation results to the end of the experimental section (Section 4.4). Furthermore, we included classification results (Section 4.1) as other commonly-used examples and studied implications for memory-efficient backpropagation.

---

> > ### Author Response · Authors · 2019-11-14
> > **Response to R3 (comment 2)**
> >
> > ....continuation of comments:
> >
> > Q: When training FlowGAN with spectral normalization, does stability come at the cost of the quality of generated samples? [More generally, what is the trade-off between expressivity and stability (e.g., also in classification)?
> > ---------------------
> > A: We added an analysis of the sample quality of MLE and ADV trained models (see Table 2 in Section 4.3). Currently, we observe a tradeoff as the FID score is lowest for the stabilized ADV-model. However, this is most likely due to limited tuning of hyperparameters, due to limited time. We will update this result in the final version.
> > ---------------------
> >
> > Q: The results would be more relevant if the architectures resembled the architectures used for invertible models used in the literature (e.g. GLOW) where we not only have coupling layers but they are interleaved with PLU linear flows.
> > ---------------------
> > A: In our classification, generative modeling, and decorrelation experiments we only used shuffling permutations (squeeze layers, see Table 1) to make the design space of INNs smaller. Future work could analyze more building blocks like PLU-flows, i-ResNets [1] or MintNet [2] blocks.
> > Lastly, we did analyze trained GLOW models that have PLU-flows in Section 4.2 by crafting non-invertible inputs.
> > ---------------------
> >
> > Furthermore, we fixed the typos you spotted.
> >
> > Thank you again for your thorough review, and we hope that you appreciate our revised manuscript.
> >
> > [1] Behrmann et al., “Invertible Residual Networks,” ICML 2019.
> > [2] Song et al., “Building Invertible Neural Networks with Masked Convolutions,” NeurIPS 2019.

---

### Official Review · AnonReviewer2 · 2019-10-22
**Official Blind Review #2**

**Rating:** 3

**Review:**

This paper analyses the numerical invertibility of analytically invertible neural networks (INN). The numerical invertibility depends on the Lipschitz constant of the respective transformation. The paper provides Lipschitz bounds on the components of building blocks for certain INN architectures, which would guarantee numerical stability. Furthermore, this paper shows empirically, that the numerical invertibility can indeed be a problem in practice.

This work is interesting and could be important to many researchers working with INNs. The worst case analysis and the corresponding table with Lipschitz bounds is useful.
However, I have some concerns regarding the experimental evaluation.
- Experiments in 4.1. nicely show that there exist non-invertible inputs for a GLOW model trained on CIFAR. But I wish the authors also considered other popular INN models and non-image datasets for this set of experiments (showing if this is also an issue in scenarios other than CIFAR/CELEBA + GLOW).
- Although the authors spend significant space in the main text and the appendix to motivate the experiments in 4.2, I cannot follow this motivation. For example, “decorrelation is a simpler objective than optimizing outputs z = F(x) to follow a factorized Gaussian as in Normalizing Flows”. Why is this is simpler, and, more importantly, why would this be an argument? Another example is “… this decorrelation objective offers a controlled environment to study which INN components steer the mapping towards stable or unstable solutions, …”. Why is this more controlled? What exactly is controlled here that is not controlled in training a an INN for, e.g., density estimation?  I am not sure if this set of experiments is any useful for determining whether numerical precision is actually problematic for posterior approximation with normalizing flows, density estimation, etc.
- the experimental sections is somewhat badly structured and makes it difficult to read. It is not clear if this paper is analysis-only or whether the authors propose a remedy. The authors write in the abstract and conclusion that they show how to guarantee invertibility for one of the most common INN architectures. After reading this, I would expect a designated experimental section which shows a fix. I suppose they refer to Additive blocks + Spectral Norm, discussed in 4.2.1. However, that reads more like a post-hoc insight (“it turns out that…” rather than “we show how“). In short, the experiments section could be much better structured.
- The paper would be greatly improved, if the authors would propose how to tackle these numerical problems. I doubt that additive coupling is “one of the most common INN architectures”. It would be nice if the authors would conduct more extensive experiments and propose solutions for other building blocks.
- I expect at least a few experiments that quantify numerical instability with multiple different random seeds (for initialization etc.).

For these reasons I vote for rejection.
I think it would be advisable to rethink the goals of the experimental evaluation, come up with a better structure, and expand at several places. E.g. (i) expand 4.1 to other architectures and data, (ii) show how this is relevant in practice (e.g. posterior inference with NFs and density estimation) and how it questions published results (currently Sec. 5), and (iii) evaluate proposed solutions.



**Experience Assessment:**

I have read many papers in this area.

**Review Assessment: Checking Correctness Of Derivations And Theory:**

I assessed the sensibility of the derivations and theory.

**Review Assessment: Checking Correctness Of Experiments:**

I assessed the sensibility of the experiments.

**Review Assessment: Thoroughness In Paper Reading:**

I read the paper at least twice and used my best judgement in assessing the paper.

---

> ### Author Response · Authors · 2019-11-14
> **Response to R2 (comment 1)**
>
> Thank you for your insightful feedback. We have updated the paper (see summary of revisions) to incorporate your comments, which we discuss below.
>
> Q: Restructure the experimental section.
> ---------------------
> A: Thank you for your comments. We restructured the experimental section based on the task to improve clarify. Please see our summary of the revision above.
> ---------------------
>
> Q: Quantify numerical instability with multiple different random seeds (for initialization etc.).
> ---------------------
> A: In preliminary experiments with different random seeds, we observe consistent behavior for each architecture. We will work on adding multiple runs of each experiment for the final version.
> ---------------------
>
> Q: The paper would be greatly improved, if the authors would propose how to tackle these numerical problems. It would be nice if the authors would conduct more extensive experiments and propose solutions for other building blocks.
> ---------------------
> A: For additive coupling blocks, we propose and study spectral normalization. This is shown in Section 4.3 to stabilize adversarially trained INNs. However, spectral normalization does not provide guarantees for affine blocks since only local bounds can be derived (see Table 1). For such model architectures, local stabilization approaches like gradient penalties could present a way to reduce numerical problems. This is certainly an interesting avenue for future work.
> Lastly, we believe that our theoretical analysis is useful to derive bounds for other architectures such as MintNet [12].
> ---------------------
>
> Q: “I doubt that additive coupling is one of the most common INN building blocks”
> ---------------------
> A: It may be true that affine blocks (non-volume preserving) are often used for density estimation, however, in other settings additive blocks are certainly heavily used. Below we provide some references [1-11] where additive blocks are used for discriminative and generative tasks.
> ---------------------
>
> Q: It is not clear if this paper is analysis-only or whether the authors propose a remedy.
> ---------------------
> A: Our main purpose for this paper has multiple aspects:
> 1) Show that instability, and hence numerical non-invertibility, is a crucial concern for INNs.
> 2) Present Lipschitz bounds as a way to understand the instabilities that occur (e.g. affine models more prone to instability than additive ones).
> 3) Show that the stability is strongly influenced by the training objective (decorrelation/Flow-GAN appears more unstable than density estimation).
> 4)Leverage the Lipschitz bounds to control instability, e.g., via spectral normalization.
> Thus our focus is on the analysis, however this analysis leads to a remedy for additive blocks, which we study in the experimental section.
> ---------------------
>
> Q:  I am not sure if this set of experiments is any useful for determining whether numerical precision is actually problematic for posterior approximation with normalizing flows, density estimation, etc.
> ---------------------
> A: Flows are essentially designed for density estimation with maximum likelihood and with the right choice of prior and training settings they appear to be reasonably stable. However, they are very popular for other purposes like memory-saving gradients, adversarial training, or to implement exotic regularizers. And in these scenarios we were able to strikingly see that invertibility quickly breaks down if no care is taken. See e.g. our new Section 4.1 on INN-based classification or Section 4.3 on generative modeling using Flow-GANs.
> ---------------------
>
> Q: Clarification of the decorrelation task:  What exactly is controlled here that is not controlled in training an INN for, e.g., density estimation?
> ---------------------
> A: At least in the given toy example (Appendix F), the decorrelation task offers both stable and unstable solutions which solve the objective perfectly. Thus, this task is controlled in the sense that there is a stable solution, which we hope to find. Interestingly, in many cases the INNs tend to choose the unstable solutions. On the other hand, for tasks like density estimation or classification we do not know if the task is solvable by stable mappings (there could be tradeoffs between stability and performance).
> ---------------------

---

> > ### Author Response · Authors · 2019-11-14
> > **Response to R2 (comment 2)**
> >
> > ...continuation of comments:
> >
> >
> > Q: Clarification of the decorrelation task: Why simpler and why is this an argument?
> > ---------------------
> > A: Decorrelation only measures linear dependencies and is thus simpler than matching a fully-factorized Gaussian as is usually done in Normalizing Flows. We believe that this serves as an interesting ablation study to density estimation. Furthermore, our experiments show that changing the objective to a less standard task such as decorrelation has a major impact on the stability of INNs. As a conclusion, limited knowledge about the factors influencing the stability of INNs could hinder their performance when using novel objectives. Hence, we chose to benchmark different architecture settings under this less standard objective, as well.
> > ---------------------
> >
> > Q: Finding non-invertible inputs for other popular INN models and non-image datasets
> > ---------------------
> > A: We included a new result on searching for non-invertible inputs for trained residual flows [13], see Appendix E. As residual flows are based on i-ResNets [14], they are by design based on certain stability bounds. In line with the theory, we thus were not able to find strong examples of non-invertible inputs.
> > Furthermore, results on non-image datasets would be of interest but are less frequently used by the mainstream invertible net literature and thus out of the scope of this article.
> > ---------------------
> >
> > Thank you again for your thorough review and hope that you appreciate our revised manuscript.
> >
> > [1] Gomez et al., “The Reversible Residual Network: Backpropagation without Storing Activations,” NeurIPS 2017 http://papers.nips.cc/paper/6816-the-reversible-residual-network-backpropagation-without-storing-activations
> > [2] Jacobsen et al., “iRevNet: Deep Invertible Networks,” ICLR 2018. https://arxiv.org/pdf/1802.07088.pdf
> > [3] Kolesnikov et al., “Revisiting Self-Supervised Visual Representation Learning,” CVPR 2019, https://arxiv.org/pdf/1901.09005.pdf
> > [4] Jacobsen et al., “Excessive Invariance Causes Adversarial Vulnerability,” ICLR 2019. https://arxiv.org/pdf/1811.00401.pdf
> > [5] Donahue & Simonyan, “Large Scale Adversarial Representation Learning,” NeurIPS 2019,  https://arxiv.org/pdf/1907.02544.pdf
> > [6] van de Leemput et al., “MemCNN: A Framework for Developing Memory Efficient Deep Invertible Networks,” ICLR 2018 Workshop. https://openreview.net/pdf?id=r1KzqK1wz
> > [7] van der Ouderaa & Worrall, “Reversible GANs for Memory-Efficient Image-to-Image Translation,” CVPR 2019. http://openaccess.thecvf.com/content_CVPR_2019/papers/van_der_Ouderaa_Reversible_GANs_for_Memory-Efficient_Image-To-Image_Translation_CVPR_2019_paper.pdf
> > [8] Brugger et al., “A Partially Reversible U-Net for Memory-Efficient Volumetric Image Segmentation,” https://arxiv.org/pdf/1906.06148.pdf
> > [9] Putzky et al., “i-RIM applied to the fastMRI Challenge,” https://arxiv.org/pdf/1910.08952.pdf
> > [10] Toth et al., “Hamiltonian Generative Networks,” https://arxiv.org/abs/1909.13789
> > [11] Hoogeboom et al., “Integer Discrete Flows and Lossless Compression,” https://arxiv.org/abs/1905.07376
> > [12] Song et al., “Building Invertible Neural Networks with Masked Convolutions,” NeurIPS 2019.
> > [13] Chen et al., “Residual Flows for Invertible Generative Modeling,” https://arxiv.org/abs/1906.02735
> > [14] Behrmann et al., “Invertible Residual Networks,” ICML 2019.

---

### Official Review · AnonReviewer1 · 2019-10-26
**Official Blind Review #1**

**Rating:** 6

**Review:**

This paper points out invertible neural networks are not necessarily invertible because of bad conditioning. It shows some cases when invertible neural networks fail, including adding adversarial pertubations, solving the decorrelation task, and training without maximum likelihood objective (Flow-GAN). The paper also shows that spectral normalization improves network stability.

I think this is a solid work. The main contribution is it points out a problem that is overlooked before, which can possibly explain some unstable behavior for training neural networks. The paper also has some study on various architectures, which sheds some light on the designing of invertible neural networks. I think this paper can be important for future researchers to design models and algorithms.

===============

Update:

After reading other reviewer's comment I agree with other reviewers that the experimental section is problematic. It seems to be unrelated with the theoretical results proposed in this paper. I think currently the experiments only make a point that invertible networks can be non-invertible in practice. But the paper has large room to improve if it has

1. A complete discussion on which invertible blocks / modeling tasks are easier to be non-invertible, and why (theoretically, and combine with direct experimental evidence)
2. A remedy (using additive coupling layer is not an acceptable one since it severely limits the modeling power)

I still think posing the problem itself is important. Thus I will still give it an accept, but lower it to a weaker score.

**Experience Assessment:**

I have read many papers in this area.

**Review Assessment: Checking Correctness Of Derivations And Theory:**

I assessed the sensibility of the derivations and theory.

**Review Assessment: Checking Correctness Of Experiments:**

I assessed the sensibility of the experiments.

**Review Assessment: Thoroughness In Paper Reading:**

I read the paper at least twice and used my best judgement in assessing the paper.

---

> ### Author Response · Authors · 2019-11-14
> **Response to R1**
>
> Thank you for your positive feedback on our work! We hope that our added results confirm your opinion on the importance of our presented ideas.
> As you pointed out, we believe that this work will help researchers to better understand current INN design choices and improve future models. In our revision, we added classification experiments which further underline the importance of stability.

---

### Author Response · Authors · 2019-11-14
**Summary of revision**

We thank the reviewers for their insightful and valuable comments. We have made several revisions to our paper which we summarize here:

1) We corrected the bound on the local Lipschitz constants of the inverse mapping of the affine coupling blocks (see Table 1 and Appendix A.2.2).

2) We restructured the experimental section based on four studied tasks (with a new section on classification):
---- Classification (Section 4.1): we show that invertible neural network (INN) classifiers can become unstable on CIFAR-10  and demonstrate implications for memory-efficient backpropagation.
---- Density estimation (Section 4.2): analysis of trained SOTA INN-based density models. Includes a new result on checking for non-invertible inputs of residual flows (Appendix E).
---- Generative modeling with adversarially trained INN models. Now includes sample evaluation using FID scores (Table 2), with a comparison to MLE-trained INN-models.
---- Decorrelation (Section 4.4): clarified the motivation.

Overall we think this structure makes our experimental analysis clearer. The new classification results demonstrate that non-invertibility is a strong concern and emphasize the need to better understand the stability of invertible networks.

Lastly, we apologize for responding late in the discussion phase. We were working hard on running new experiments and thoroughly revising our manuscript.

---

### Decision · Program_Chairs · 2019-12-19

**Decision:**

Reject

**Comment:**

This submission analyses the numerical invertibility of analytically invertible neural networks and shows that analytical invertibility does not guarantee numerical invertibility of some invertible networks under certain conditions (e.g. adversarial perturbation).

Strengths:
-The work is interesting and the theoretical analysis is insightful.

Weaknesses:
-The main concern shared by all reviewers was the weakness of the experimental section including (i) insufficient motivation of the decorrelation task; (ii) missing comparisons and experimental settings.
-The paper clarity could be improved.

Both weaknesses were not sufficiently addressed in the rebuttal. All reviewer recommendations were borderline to reject.